# Presynaptic gating of monkey proprioceptive signals for proper motor action

Saeka Tomatsu[1,2,3], GeeHee Kim[1,4,,5], Shinji Kubota[1] & Kazuhiko Seki [1,3,4] ✉

Our rich behavioural repertoire is supported by complicated synaptic connectivity in the central nervous system, which must be modulated to prevent behavioural control from being overwhelmed. For this modulation, presynaptic inhibition is an efficient mechanism because it can gate specific synaptic input without interfering with main circuit operations. Previously, we reported the task-dependent presynaptic inhibition of the cutaneous afferent input to the spinal cord in behaving monkeys. Here, we report presynaptic inhibition of the proprioceptive afferent input. We found that the input from shortened muscles is transiently facilitated, whereas that from lengthened muscles is persistently reduced. This presynaptic inhibition could be generated by cortical signals because it started before movement onset, and its size was correlated with the performance of stable motor output. Our findings demonstrate that presynaptic inhibition acts as a dynamic filter of proprioceptive signals, enabling the integration of task-relevant signals into spinal circuits.

The proprioceptive sensory signal has a fundamental role in coordinating body movement[1,2]. It arises from a population of mechanoreceptors primarily located in the skeletal muscles and tendons and is relayed to the central nervous system (CNS) through muscle and tendon afferent projections to the spinal cord. Then, it shapes proprioception (sense of body position) by ascending to the cerebral cortex, and it also generates muscle activity through the spinal reflex circuits. The rich behavioural repertoire of animals stimulates these ubiquitous mechanoreceptors and activates CNS neurons in various fashions, so the sensory input from muscle afferents to the spinal cord must be modulated appropriately to prevent movement control from being easily overwhelmed. However, how the CNS regulates and integrates these signals to shape proper proprioception and reflex action is unknown. For example, simple joint movements generate the activity of spindle afferents from agonistic[3] and antagonistic[4] muscles. However, how the afferent signal is regulated to make a clear sense of

joint angle and the autogenic and reciprocal reflex output for movement control has not been established.

The efficacy of synaptic transmission between primary afferents to CNS neurons can be modulated by presynaptic inhibition (PSI). PSI occurs throughout the CNS, e.g. spinal cord[5], ventral tegmental area[6], hippocampus[7], cerebellum[8], visual cortex[9], and brainstem[10]. In the spinal cord, where PSI was originally discovered[11], GABAergic interneurons[12,13] form axo-axonic contacts at the intraspinal terminals of primary sensory afferents. The release of GABA at these presynaptic contacts reduces the release of afferent transmitters and thus suppresses synaptic transmission. PSI at primary afferents is generated by input from descending pathways[14,15] as well as from homonymous and heteronymous primary afferents[15,16]. However, as with PSI at other locations in the CNS, the behavioural relevance of spinal cord PSI was unknown until recently.

Previously, we reported evidence that PSI modulates cutaneous input to the primate spinal cord during normal voluntary movements[17,18].

[1]National Institute of Neuroscience, National Center of Neurology and Psychiatry, Kodaira Tokyo, Japan. [2]Division of Behavioral Development, Department of System Neuroscience, National Institute for Physiological Sciences, National Institutes of Natural Sciences, Okazaki Aichi, Japan. [3]Department of Physiological Sciences, School of Life Science, The Graduate University for Advanced Studies (SOKENDAI), Hayama, Kanagawa, Japan. [4]Division of Behavioral Development, Department of Developmental Physiology, National Institute for Physiological Sciences, National Institutes of Natural Sciences, Okazaki Aichi, Japan. [5]Present address: Graduate School of Arts and Sciences, The University of Tokyo, Komaba, Tokyo, Japan. ✉e-mail: seki@ncnp.go.jp

In monkeys performing wrist movements, monosynaptic input from forearm cutaneous afferents to spinal interneurons was suppressed during active movements. This suppression occurred in the task epoch when postsynaptic excitability was increased; thus, it could not be explained by a postsynaptic inhibitory mechanism. Consequently, we proposed that their suppression was generated by a PSI mechanism. To support this proposal further, we established a method to assess the level of primary afferent depolarization (PAD) of cutaneous afferents, which reflects the GABAergic depolarization underlying PSI at primary afferent terminals[19,20], by modifying the excitability testing technique[21], which was described in our subsequent report[18]. In brief, the level of PAD could be reflected by the size of antidromic volleys (ADVs) elicited by intraspinal stimulation, which were observed at peripheral nerves. By using this method, we found increased PAD during dynamic movements, and concluded that the cutaneous afferent input to the spinal cord was indeed suppressed by PSI. We further found that PSI is driven by descending motor commands, presumably to suppress task-irrelevant cutaneous signals. Following this report, a number of studies reported a refined mechanism[13,22,23] and the clinical relevance[24–26] of PSI for normal behaviour in experiments using rodents or human patients.

Importantly, PSI of cutaneous afferents is not movement specific; it suppresses cutaneous afferent feedback during agonistic and antagonistic movements[17,18]. This global facilitation of PSI seen in the cutaneous afferent input would be problematic if it is also implemented in the proprioceptive sensory system, because proprioceptive afferent activity is highly specific to movement direction[4]. Thus, this movement selectivity in proprioceptive afferent activity could be blurred if it is suppressed by PSI irrespective of movement context, as we found for the cutaneous afferent input. To address this issue, we compared the monosynaptic input to the spinal cord from muscle and cutaneous afferents and examined if muscle input was modulated differentially to cutaneous input during movement[27]. We found that, at the first relay neuron, the response to the input from muscle afferents during agonistic movements was generally facilitated, which was in clear contrast to the cutaneous responses[17]. Importantly, we could not determine the

underlying synaptic mechanism of this response facilitation because it was observed in the epoch when the general excitability of neurons was also enhanced. In this case, response modulation could represent either the presynaptic or postsynaptic modulatory mechanism or both.

Therefore, in the present study, we examined whether proprioceptive input during voluntary movements is modulated by PSI. We evaluated the moment-by-moment dynamics of PAD in awake monkeys performing wrist flexion-extension movements using a comparable method to that established in our previous study on cutaneous afferents[17,18]. We found evidence of PAD suppression during agonistic movements that suggests the facilitation of the muscle afferent response in spinal neurons[27] is generated by PSI. We also observed that PSI was enhanced during antagonistic movements. Moreover, PSI during agonistic and antagonistic movements could be generated by cortical signals because it started before movement onset, and the size of PSI was correlated with the performance of stable motor output. Our results demonstrate that PSI acts as a dynamic filter of proprioceptive signals, enabling the integration of only task-relevant signals into spinal circuits for proper motor action.

## Results

### PAD modulation during a motor task

Two monkeys learned to perform wrist extension-flexion movements (Fig. 1a, b). During this task, we applied repetitive microstimulation (10 Hz, <50 µA; without any observable muscular twitches) to lower cervical intraspinal sites (C5–C8, Fig. 1c) and recorded the responses from the deep radial (DR) nerve, which includes proprioceptive afferents from wrist extensor muscles and their tendons (Fig. 1d, e). We observed 77 significant antidromically conducted responses (Fig. 1e, ADVs) evoked by stimulating 36 intraspinal sites within the dorsal horn. The conduction velocities of these ADVs ranged from 44.3 to 79.1 m/s (mean ± standard deviation [SD]: 65.4 ± 8.1 m/s), indicating that the axon terminals of group I and/or II primary afferent fibres had been stimulated[28]. See Supplementary Table 1 for the full profiles of each ADV. The response to each microstimulation during wrist

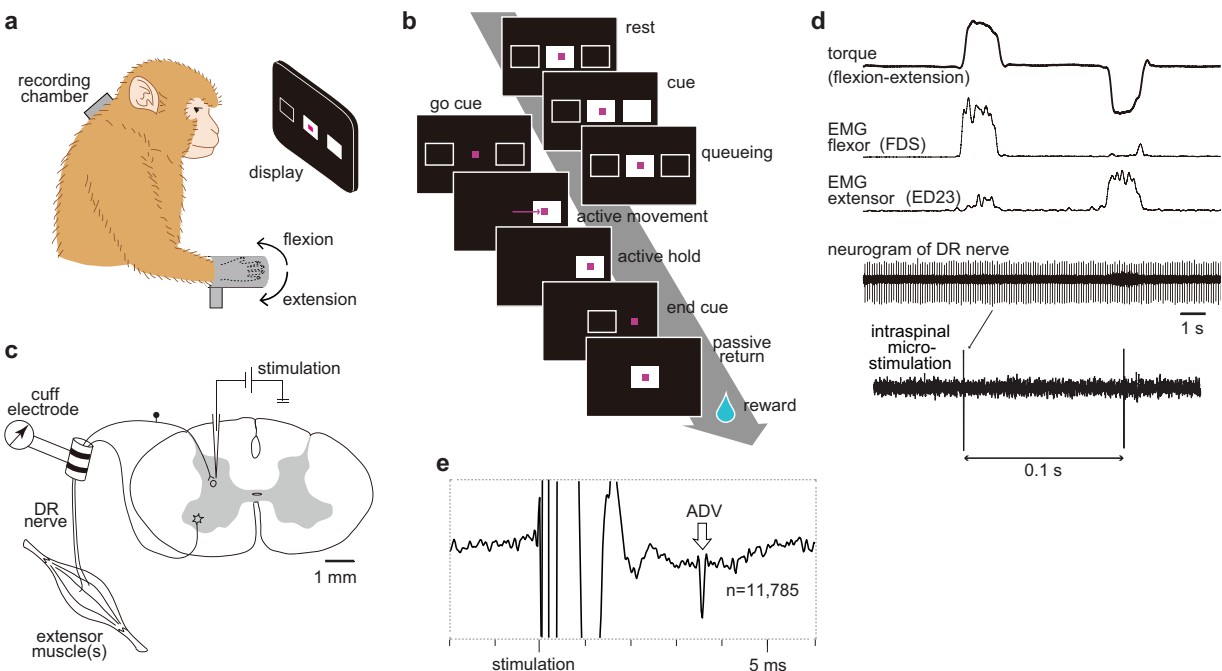

**Fig. 1 | Experimental setup and examples of antidromic volleys. a** Monkey with a recording chamber on the cervical spinal cord performing the movement task with visual cues. Adapted with permission of Kazuhiko Seki, from Principles of Neural Science, Kandel, Erik R, 5th Edition, 2013; permission conveyed through Copyright Clearance Center, Inc. **b** Task sequence for an extension trial. Magenta cursor, feedback of wrist torque. **c** Electrophysiological recording setup. **d** An example of wrist torque (upwards reflects flexion), electromyography (EMG) traces from a flexor and an extensor, and a neurogram from the deep radial (DR) nerve. ED23, extensor digitorum-2,3; FDS, flexor digitorum superficialis. **e** A typical example of antidromic volleys (ADVs, #17 in Supplementary Table 1).

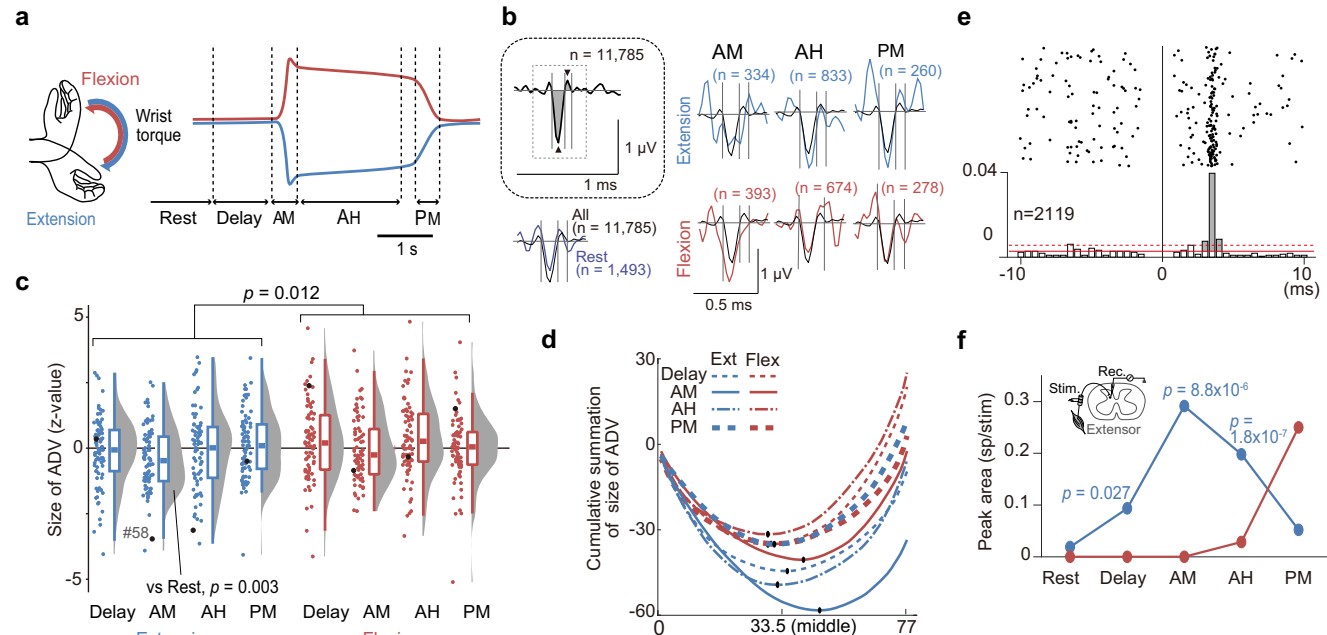

**Fig. 2 | Modulation of presynaptic inhibition during different task epochs. a** An illustration of the wrist torque and task epochs. Blue; extension. Red; flexion. The colour code is consistent for all panels. AM, Active Movement; AH, Active Hold; PM, Passive Return Movement. **b** Measurement of antidromic volley (ADV) area and its modulation during the movement task. Data from the same ADVs shown in Fig. 1e are displayed in this figure. Inset, ADVs obtained using all intra-spinal micro-stimulations (*n* = 11,785) while the monkeys performed the task. Prominent negative (rectangle) and positive (inverted rectangle) peaks were found and the onsets and offsets (vertical grey lines, baseline-crossing points) were detected. Grey-shaded areas indicate the area measured for the ADVs. Bottom left, and top and bottom right, Averaged ADV area modulation in different task epochs. Waveforms obtained with a global average (same as in inset) are overlaid in each panel (black). Vertical grey lines indicate the ranges of area calculation, with the same timing as

shown in inset. **c** Density and box-whisker plots of the size of the ADVs (*n* = 77 ADVs) for each behavioural epoch. Black dots indicate the ADV elicited from an intraspinal site (site #58) where the activity of the first-order spinal neuron was successfully recorded as illustrated in (**e** and **f**). **d** Cumulative summation of the size of 77 ADVs. Black dots indicate the minimum value of each curve. **e** Raster plot and peristimulus time histogram of a single first-order spinal neuron to a deep radial (DR) afferent. Zero indicates the timing of stimulation. The peristimulus time histogram summarizes the firing profile of the neuron in response to stimulation, with a 0.5-ms bin. Grey area indicates the range calculated as the response probability of this neuron. **f** Peak area of the neuronal response shown in (**e**) in each behavioural epoch. *P* values are from two-tailed binomial tests compared with Rest with Bonferroni's correction (correction size = 4). Source data are provided as a Source data file.

movements was normalized to those observed in the canonical period (Rest), compiled independently for each behavioural epoch, and then used to evaluate changes in PAD, as the level of PAD positively correlated with the size of ADVs elicited by intraspinal stimulation[21]. These analyses were performed on perfect trials, as defined by the appropriate direction (flexion or extension), timing, and duration of the executed movements for all behavioural epochs. For the background and rationale of this analysis, please refer to Supplementary Note 1.

First, we found the distinct modulation of PAD (Fig. 2b–d) depending on the behavioural epoch (Fig. 2a). In the wrist extension trials, PAD suppression occurred during the Active Movement (AM) epoch (Fig. 2c, mean = −0.44, df = 76, t = 3.06, uncorrected *p* = 0.003, paired two-tailed t-test with Bonferroni's correction, correction size = 4, compared with Rest), suggesting that afferent input from extensor muscles is facilitated during dynamic wrist extension. The other epochs did not differ from the Rest period (Delay, mean = −0.12, t = 0.81, *p* = 0.42; AH, mean = −0.07, t = 0.41, *p* = 0.68; PM, mean = 0.11, t = 0.79, *p* = 0.43; df = 76, *p*-values are uncorrected, paired two-tailed t-test with Bonferroni's correction, correction size = 4). In the wrist flexion trials, it was noteworthy that the sole negative mean was observed in the AM epoch, in common with the wrist extension trials, although none of the epochs differed from the Rest period (Delay, mean = 0.26, t = 1.44, *p* = 0.13; AM, mean = −0.03, t = 0.21, *p* = 0.83; AH, mean = 0.33, t = 1.95, *p* = 0.05; PM, mean = 0.04, t = 0.26, *p* = 0.79, df = 76, *p*-values are uncorrected, paired two-tailed t-test with Bonferroni's correction, correction size = 4). The specificity of the AM epoch was also detected in the cumulative summation plot of PAD size (Fig. 2d,

blue and red solid lines); They were clearly separate from the other traces of respective movement directions and biased to negative values. These results led us to conclude that PSI of the muscle nerve input during voluntary movements has a non-directional characteristic, i.e. it facilitates proprioceptive input during dynamic movements.

Next, we compared PAD modulation between the different movement directions. We found that PAD in the flexion trials was larger than in the extension trials (Fig. 2c, mean = −0.13 and 0.19, respectively, df = 306, t = 2.51, *p* = 0.012, paired two-tailed t-test, compared between the aggregated data of the flexion and extension trials), suggesting that afferent input from extensor muscles is suppressed more during the flexion trials than during the extension trials. Such movement direction-related modulation was also reported in previous studies using the wrist extension-flexion task, e.g. the reciprocal activity of premotor interneurons in the spinal cord[29] and the reciprocal activity of neurons in the primary motor area[30].

Then, we examined the polarity of the directional modulation of PAD. In this analysis, we exclusively used the static task epochs (i.e. the Delay and Active Hold [AH] epochs), which comprised the largest part of each trial (75–82%) and exhibited a consistent feature within each movement direction (Fig. 2d, thin dotted and dashed lines). The dynamic task epochs were excluded because we had already identified direction-independent modulation during these epochs. We found that the aggregated PADs of the Delay and AH epochs was significantly larger than in the Rest period in the flexion trials (mean = 0.29, df = 153, t = 2.46, uncorrected *p* = 0.02, paired two-tailed t-test with Bonferroni's correction, correction size = 2). We did not observe a comparable

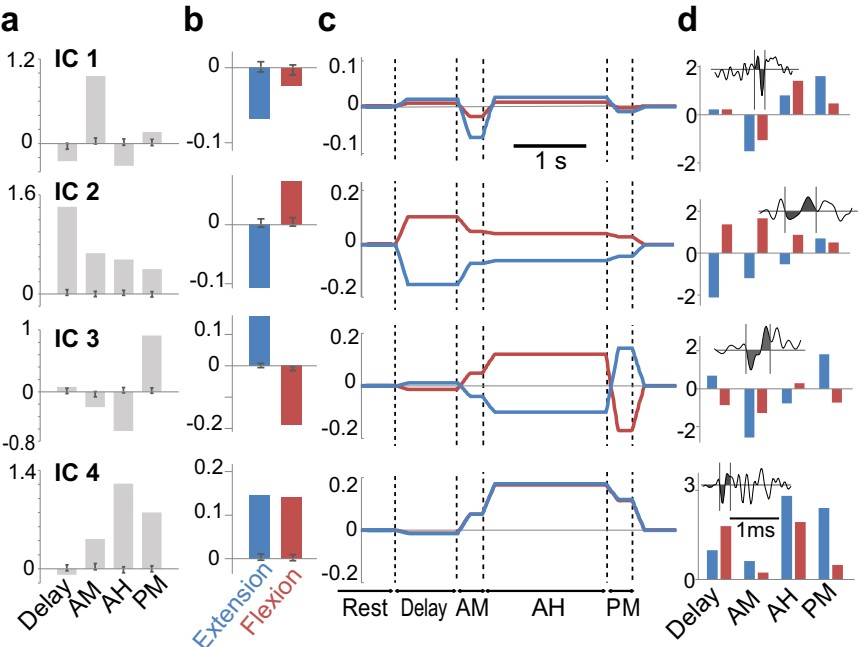

**Fig. 3 | Independent component analysis and presumed modulation of presynaptic inhibition. a** Four independent components (ICs) for the behavioural modulation of antidromic volleys (ADVs). Dark grey vertical lines indicate confidence intervals calculated by bootstrap datasets ($n = 1000$ times). AH, Active Hold; AM, Active Movement; PM, Passive Return Movement. **b** Median value of the weight of each IC (mixing weight, $n = 77$ ADVs). Dark grey vertical lines indicate confidence intervals calculated by bootstrap datasets ($n = 1000$ times). **c** Presumed pattern of ADV modulation, illustrated based on the extracted ICs (**a**) and mixing weights (**b**). **d** Example of four ADVs exhibiting pattern modulation of their size across different behavioural epoch and their waveforms (inset). Source data are provided as a Source data file.

modulation in the extension trials (mean = −0.10, df = 153, t = 0.84, uncorrected $p = 0.40$, paired two-tailed t-test with Bonferroni's correction, correction size = 2, compared with Rest), but it was significantly smaller than in the flexion trials (df = 153, t = 2.30, $p = 0.02$, paired two-tailed t-test), as indicated by the grand average of all epochs. Therefore, we can conclude that afferent input from extensor muscles was suppressed during the flexion trials, but not during the extension trials.

These principle characteristics of epoch-dependent PAD modulation were consistent in both monkeys (Supplementary Fig. 1a, b), and did not change throughout all recording days (epoch × day two-way analysis of variance, no significant interaction, for Monkey Y: extension, df = 48, F = 1.16, $p = 0.21$; flexion, df = 48, F = 0.67, $p = 0.96$, for Monkey O: extension, df = 39, F = 0.76, $p = 0.87$; flexion, df = 39, F = 1.03, $p = 0.42$). PAD modulation could not be ascribed to nonphysiological factors that might change the size of the ADVs (Supplementary Note 2). The suppression of PAD was found in the same phase (AM of extension, Fig. 2b–d) as in our previous report on the increased response probability of spinal first-order interneurons from DR afferents[27], strongly suggesting that PAD modulation affects the excitability of relay neurons; increased excitability by PAD suppression and decreased excitability by PAD facilitation. Indeed, we found an interneuron exhibiting response facilitation during the AM epoch of extension (Fig. 2e, f) at the same intraspinal site where predominant PAD suppression was observed during the same phase (Fig. 2c, black dots). Therefore, we concluded that afferent input from extensor muscles was facilitated (by decreased PSI) during the AM epoch of the extension trials, and which was suppressed (by increased PSI) during most epochs of the flexion trials. Overall, we found reciprocal modulation of PSI depending on movement direction.

## Components of PAD modulation
Previous studies on anaesthetized animals or reduced preparations reported that various nerve-input combinations increase or decrease

PAD[19], suggesting that PAD modulation stems from multiple sources. These sources might not be represented in the total average (Fig. 2) because they could operate PAD in parallel during voluntary movements and thus their effect might be offset to each other. To compensate for this disadvantage, we extracted the components underlying the observed task-dependent modulation by independent component analysis (ICA), which hypothesized multiple causes to make temporal modulations of PSI at each task epoch and movement (Fig. 2b–d). This analysis yielded four ICs (IC1–4, Fig. 3a) and mixing weights (Fig. 3b), which together form the potential basis of PAD modulation during flexion and extension movements (Fig. 3c). The IC values and medians of mixing weights exceeded the 95% confidence interval (Fig. 3a, b, dark grey lines), suggesting they are independent of noise and, thus, significant. Individual ADVs whose modulation indicated higher weight values for each IC are also illustrated in Fig. 3d.

A feature of IC1 was the transient disinhibition (decreased PAD) of afferent input during the AM epoch in the flexion and extension trials, while a common feature of IC2, IC3, and IC4 was increased PAD during the flexion trials (Fig. 3c). For IC2, PAD began to increase during the preparation period (Delay), indicating that the IC2-driven enhancement of PAD is triggered by descending commands[14,15]. Interestingly, this modulation was reciprocal, i.e. facilitation for flexion and suppression for extension, which could be related to a top-down sensory-gating mechanism[31–33]. In contrast, for IC3 and IC4, PAD began to increase predominantly during the AM epoch and both showed higher weight values during the Passive Return Movement (PM) epoch, which would be expected if this modulation is generated by reafference signals from the periphery.

The temporal features of these components suggest that the PSI-induced modulation of sensory gain in the spinal cord during voluntary movements might be generated by a combination of descending commands and reafference signals, rather than simply by a single source. Functionally, we can summarize these results as follows. When

a muscle acts as an antagonist, i.e. passively lengthening flexion movements in this study, the afferent input is suppressed by increased PSI. This suppression is sustained throughout the task (IC2, 3, and 4), and a portion of it is generated by descending motor commands (IC2). Conversely, afferent input from the muscle is transiently facilitated by decreased PSI (IC1) during the AM epoch, regardless of whether the muscle works as an agonist or antagonist. We suggest that proprioceptive sensory input during voluntary movements might be processed using a combination of these (two) functional components. As a result, the input during the AM epoch is highly biased towards agonistic movements (Fig. 2) because the decreased PSI that would have been provided from IC1 was offset by the increased PSI (IC2–4) during the antagonistic AM period.

## Descending commands modulate PAD in two ways

As described above, the enhancement of PAD in IC2 must have a descending origin because it began during the Delay period, before the movement was initiated. The source of the transient PAD suppression in IC1 was less immediately clear because we could not determine the accurate timing of the onset of PAD suppression relative to the onset of movement within the resolution of the epoch-based analysis shown in Fig. 2. Thus, we compared the temporal patterns of the ADV areas by aligning them to either electromyography (EMG) onset (Fig. 4a) or torque onset (Fig. 4b). The rationale here is that the ADVs should be aligned more sharply to EMG onset if they are generated by descending commands, and to torque onset if they are generated by force or by displacement feedback as a consequence of movement.

Analysis revealed more prominent transient PAD suppression (as shown in IC1) when the ADVs were aligned to EMG onset (Fig. 4a) than when they were aligned to torque onset (Fig. 4b). Furthermore, such suppression was not reproduced when they were aligned to EMG offset (i.e. termination of the descending command, Fig. 4c) or torque offset (onset of the second change in displacement, Fig. 4d). Therefore, we conclude that the transient PAD suppression in IC1 was derived primarily from descending commands for initiating movement, perhaps from collaterals of motor commands that activate motoneurons, but not from sensory reafference.

While the characteristics of the AM epoch in IC1 were also reflected in the results of averaging analyses (Figs. 2c, 4a, significant difference with the Rest period), those of the Delay epoch in IC2 were less represented (Figs. 2c, 4a, no significant difference with the Rest period). These results suggest that a larger population of ADVs is modulated by the descending source for IC1, but a smaller subpopulation is affected by the descending source for IC2. These two types of PAD modulation by descending commands can reasonably explain our previous findings demonstrating that first-order interneurons from the DR nerve, the postsynaptic cells of afferent terminals, exhibit increased excitability during active wrist extensions that starts earlier than EMG onset, and decreased excitability during wrist flexion[27]. The correspondence of the current results and our previous ones, together with an example of PAD suppression and response facilitation at the same spinal site (Fig. 2f), suggest that the PAD modulation observed in this study is sufficiently large to affect the activity of postsynaptic spinal interneurons. Therefore, we set out to determine whether this modulation might also affect task performance.

## PAD associated with sustained motor output

To compare PAD modulation among different task performances, we first extracted the successful trials and error trials, which were short hold trials in which the monkeys did not hold their movement long enough to be defined as a success (Supplementary Table 2), from all trials in which movement onset in the correct direction was detected. We used two windows to assess ADV area: a dynamic task epoch

(300 ms from EMG onset; shading in Fig. 5a upper panel) and a static task epoch (combination of Delay and AH; assessing comprehensive modulation in each trial, shading in Fig. 5a lower panel). These assessment windows included all short hold trials for the dynamic task epoch, and a few short hold trials that were sustained until the AH epoch for the static task epoch. Then, we sorted all analysed trials according to the average ADV area evoked within each assessment window. Comparisons of multiple task performance measures (Supplementary Fig. 3) between trials with relatively larger (largest third of the population) or smaller (smallest third) ADVs revealed two behavioural measures showing differences between these trials (Fig. 5, see also Supplementary Fig. 3).

First, during the extension task when PAD was transiently suppressed (Fig. 5b, d, e), there were more erroneous trials when the ADVs evoked during the AM epoch were smaller (more suppressed PAD, Fig. 5d, df = 76, t = 2.54, $p$ = 0.013, two-tailed paired t-test). As the erroneous trials frequently lacked persistent extensor EMG activity (black in Fig. 5b), we hypothesized that the preceding larger PSI might assist in maintaining torque by sustaining extensor activity. Indeed, we found that extensor EMG amplitude during the larger ADV trials was significantly larger during the subsequent AH epoch (extensor carpi ulnaris, df = 76, t = 2.08, $p$ = 0.041; extensor digitorum communis, df = 76, t = 2.22, $p$ = 0.030, two-sample paired t-test, Fig. 5e, lines with open circles). Second, during flexion when PAD facilitation was sustained (Fig. 5c, f, g), we found no significant correlation between ADV size and the success ratio (Fig. 5f, df = 76, t = 1.63, $p$ = 0.11), but the flexor EMG amplitude was significantly larger in the AH epoch (flexor carpi radialis, df = 76, t = 2.25, $p$ = 0.027; flexor digitorum superficialis, df = 76, t = 2.99, $p$ = 0.004, two-sample paired t-test, Fig. 5g, lines with open circles), as was observed in the representative example (Fig. 5c). Since this result was not replicated when we repeated a comparable analysis of the ADVs elicited in the Delay and AH epochs separately (Delay, df = 76, t = 1.23, $p$ = 0.22; AH, df = 76, t = 0.53, $p$ = 0.60, two-tailed paired t-tests, Supplementary Fig. 4), we could not ascribe this observation to the unique features of either the Delay or AH epoch. Rather, this result represents a characteristic of the flexion trials and suggests that the larger PAD throughout a trial helps to sustain EMG activity. Overall, these results suggest that larger PAD before and during movements could help to sustain the static motor output for agonistic and antagonistic movements.

## Relationship between PAD strength and motor output oscillations

Fink et al.[13] demonstrated oscillatory forelimb movements in mice in which PSI was genetically ablated. To determine if comparable motor oscillations occurred in our trials with reduced PSI, we computed frequency spectrograms and auto-correlograms for torque and EMG signals during the period following movement onset, and compared them in trials with either smaller or larger ADVs in the static task epoch (Fig. 6). There was no significant difference in the frequency spectrograms for either torque or EMG between the smaller and larger ADV trials (torque in extension, df = 76, ts = 0.0051–2.42, $p$s <0.05; EMG in extension, df = 76, ts = 0.027–4.11, $p$s <0.05; torque in flexion, df = 76, ts = 0.0026–2.35, $p$s <0.05; EMG in flexion, df = 76, ts = 0.0011–2.98, $p$s <0.05, paired two-tailed t-test with Bonferroni's correction, correction size = 30), nor was there any difference in the mean lag or mean correlation coefficient of the second peak of the torque auto-correlograms (lag in extension, df = 76, t = 0.51, $p$ = 0.61; correlation coefficient in extension, df = 76, t = 0.24, $p$ = 0.81; lag in flexion, df = 76, t = 1.35, $p$ = 0.18; correlation coefficient in flexion, df = 76, t = 0.29, $p$ = 0.77, two-tailed t-test). Furthermore, the correlation in the EMG auto-correlogram was not high (only 7.5% of trials indicated a correlation coefficient >0.3). These results imply that the PAD modulation observed in this study never triggered oscillations of motor output. We suggest that in normal animals, behavioural control through PAD

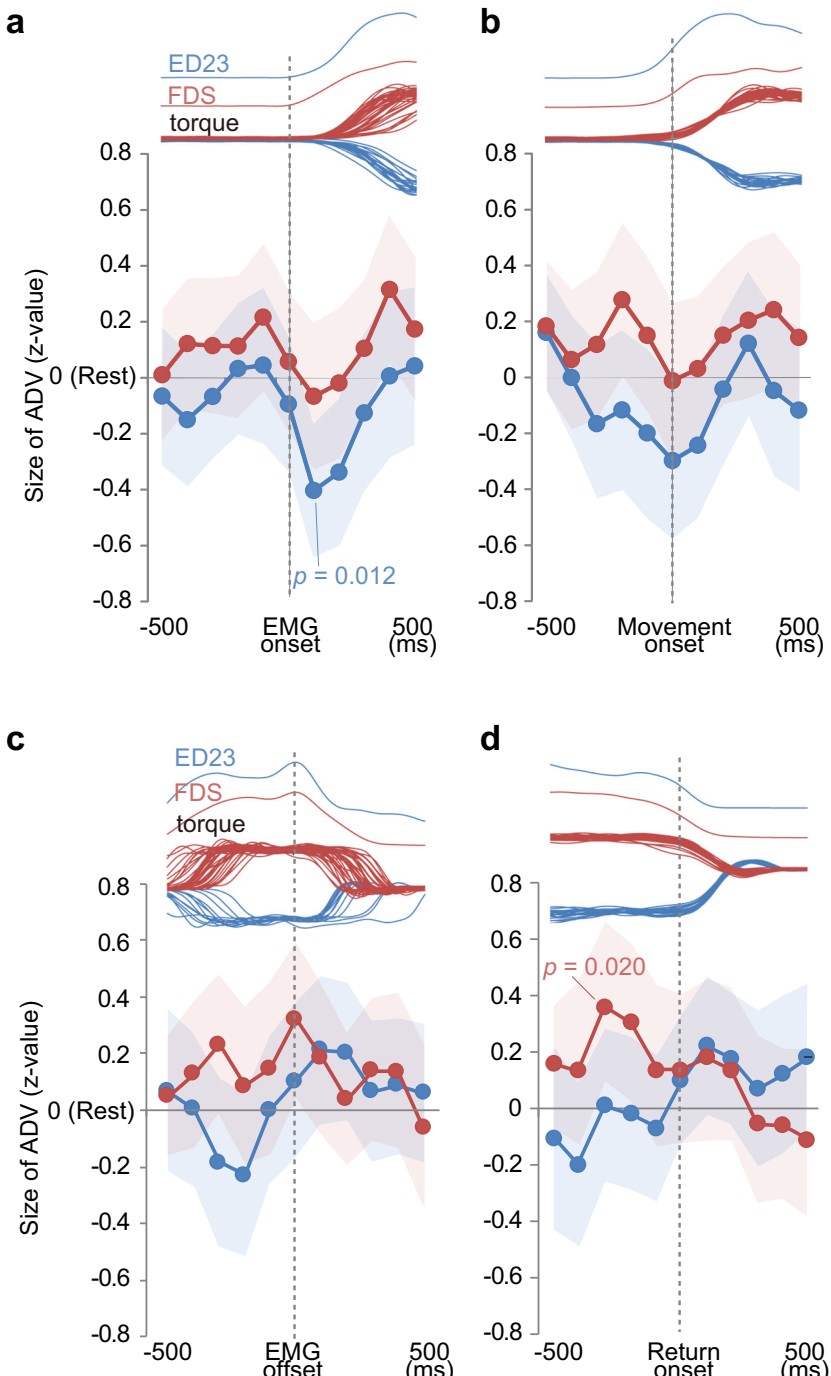

**Fig. 4 | Temporal modulation of antidromic volleys (ADVs) by motor commands or their consequences. a–d** Upper traces, averaged pre-processed electromyography (EMG) traces for the extensor digitorum-2,3 (ED23) in the extension trials (top) and the flexor digitorum superficialis (FDS) in the flexion trials (middle), and smoothed torque traces (bottom) from 30 successful extension (blue) and flexion (red) trials. Lower traces, average ($n = 77$ ADVs, circles and lines) and 95% confidence interval (shaded area) of normalized ADV area. *P* values are from two-tailed t-test compared with 0 (Rest) using Bonferroni's correction (correction size = 11). Dotted lines indicate events for data alignment: onset of EMG burst (**a**), onset of torque (**b**), offset of EMG burst (**c**), and offset of wrist torque (**d**). Source data are provided as a Source data file.

## Discussion

This study provides evidence that PSI of muscle afferents is dynamically modulated during normal voluntary movements. Although one might reasonably expect sensory attenuation of muscle afferent signals[34]—a phenomenon in which volitional motor commands attenuate predicted sensory feedback[35]—we found that PSI was transiently suppressed during active movements, not enhanced. This suggests that PSI modulation provides not only sensory attenuation but also amplification[36,37] during voluntary movements.

Our results showed that descending commands can enhance and suppress the level of PSI directed towards muscle afferents, depending on the role of the host muscles in the context of ongoing movements, i.e. agonist or antagonist (Fig. 7). For instance, assuming there is co-

modulation is achieved via a sufficient safety margin that protects against oscillatory motor output.

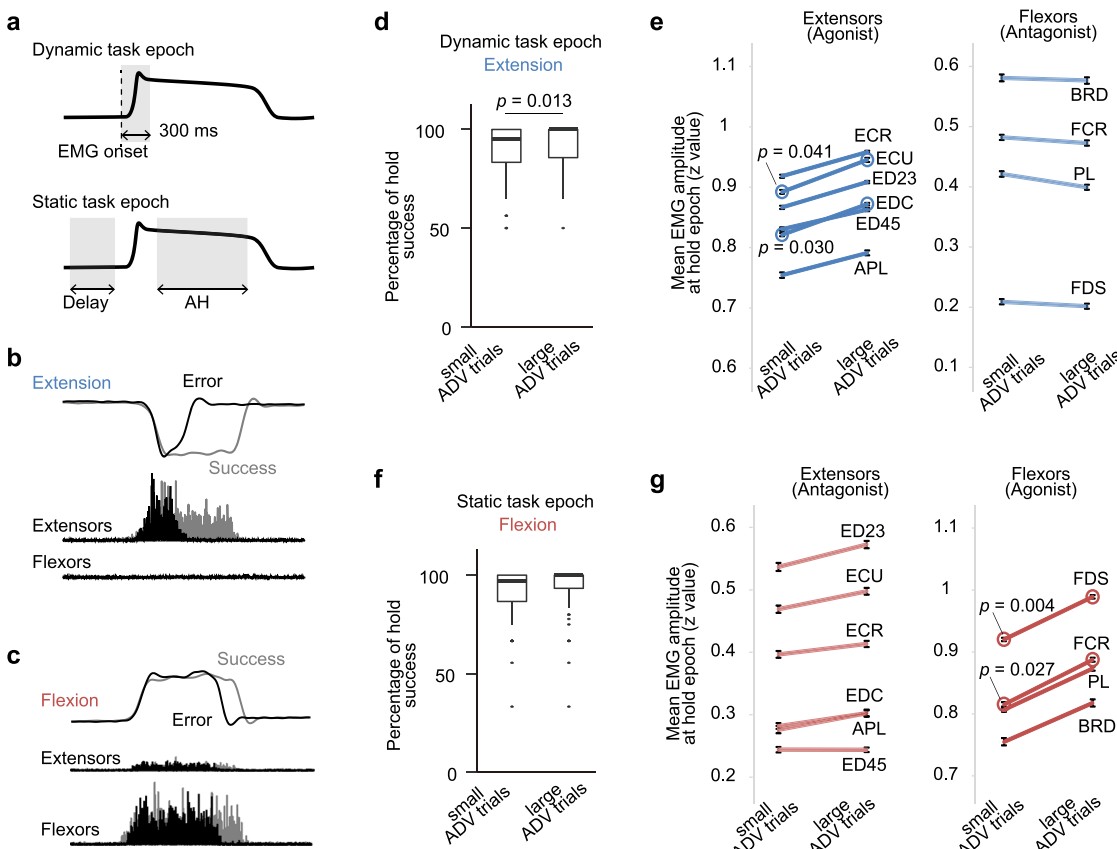

**Fig. 5 | Relationship between antidromic volley (ADV) modulation and task performance. a** Assessment windows (shaded areas) for computing the mean ADV area for each trial. **b** Sample error (black) and successful (grey) trials of wrist torque and EMG during extension movements. **c** Sample error (black) and successful (grey) trials of wrist torque and EMG during flexion movements. **d** Hold success ratio in the extension trials classified by the mean ADV in the dynamic task epoch. *P* value is from two-tailed paired t-test (*n* = 77 ADVs). **e** Means and standard errors of the EMG amplitude of individual wrist extensor and flexor muscles during the Active Hold (AH) epoch of extension trials classified by the mean ADV in the dynamic task epoch. Circle, *p* values are from two-tailed paired t-test between large

and small ADV trials (*n* = 77 ADVs). APL, abductor pollicis longus; BRD, brachioradialis; ECR, extensor carpi radialis; ECU, extensor carpi ulnaris; ED23, extensor digitorum-2,3; ED45, extensor digitorum-4,5; EDC, extensor digitorum communis; FCR, flexor carpi radialis; FCU, flexor carpi ulnaris; FDS, flexor digitorum superficialis; PL, palmaris longus; PT, pronator teres. **f** Same as for panel **d**, but the data were from flexion trials classified by the mean ADV in the static task epoch. **g** Same as for panel (**e**), but the data were from flexion trials classified by the mean ADV in the static task epoch. **d**–**g** are also illustrated in Supplementary Fig. 3. Source data are provided as a Source data file.

activation of alpha and gamma motoneurons[38], voluntary motor commands activate gamma motoneurons to maintain proprioceptive sensitivity during the agonistic action of muscles. The reduced PSI during this agonistic period (Fig. 2b–d) may assist the gamma drive to facilitate further proprioceptive feedback to the spinal cord (Fig. 2e–f), which is crucial for shortening muscles when performing dynamic movements[39]. Conversely, during the antagonistic action of muscles, the enhanced PSI suggests that feedback from lengthening muscle afferents is less informative or potentially distractive. Possibly, a sensory prediction signal generated by an internal model[40] can be more informative than the actual feedback from antagonists, or the gain of muscle spindles could be too low to encode an informative signal because of the presumably lower activity of gamma motoneurons. An intriguing question for future study is how this parallel and somehow redundant operation of proprioceptive gain control[41,42] by the gamma and PSI systems is organized in voluntary movements.

ICA suggested that this reciprocal modulation of PSI in the flexion and extension tasks was already observed in the Delay period, i.e. a period for preparing future movements (IC2 in Fig. 3). This reciprocal modulation during the Delay period was less reflected in simple analysis using averaged data (Figs. 2c, 4a), suggesting it may represent the characteristics of a smaller subpopulation of ADVs. Since no overt motor action has been initiated yet in this period, the

relevance of PSI modulation should be different from that during the movements discussed above. It is widely known that neurons in the primate sensori-motor cortex[43–45] show preparatory activity for upcoming movements. It is likely that the corticospinal input to these preparatory signals could be the source of PSI modulation. Then, why does the cortical motor preparatory signal modulate PSI in a reciprocal way, similar to the one found during movement? We suggest that the suppression of PSI before agonistic extensor movements (Figs. 2, 3, and 4) aids proper movement initiation. In an earlier report in human subjects using a reflex-testing battery, Hultborn et al.[46] suggested that the Ia input from contracting muscles is facilitated by decreased PSI at movement onset. Since this facilitation is reported at the very onset of EMG activity, we assume that the descending command suppressed PSI even before the moment of movement initiation as a set or priming signal for an upcoming movement. If this is the case, the gain of proprioceptive input in this preparatory period should be set at below, but close to, the recruitment threshold of agonistic motoneurons by PSI suppression, probably together with the modulation of the excitability of gamma motoneurons. Supporting this suggestion, we found the facilitation of DR-evoked monosynaptic responses during the Delay period (Fig. 2e, f), which is consistent with our previous report[27]. Conversely, the potential role of PSI facilitation observed before antagonistic flexor movements

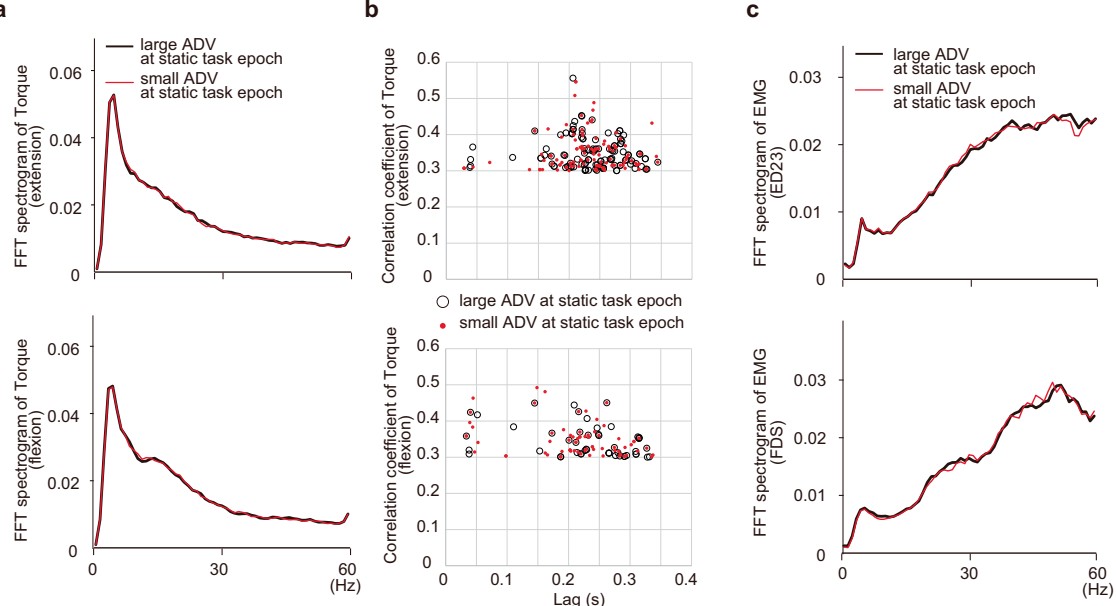

**Fig. 6 | Relationship between antidromic volley (ADV) area and potential oscillatory motor output. a** Grand average of the power spectrogram (*n* = 77 ADVs) for wrist torque during extension (top) and flexion (bottom). The fast Fourier transformation (FFT) spectrogram for each trial was categorized by ADV area during the static task epoch and averaged for each ADV. Red and black lines, averages of the small and large ADV trials, respectively (*n* = 77 ADVs). The torque signals just after movement onset (0–1 s after movement onset) were analysed. A small peak in the range of physiological oscillations was noted (5–20 Hz), but there was no significant difference between the large and small trials at any frequency (two-tailed paired t-tests with Bonferroni's correction, correction size = 60, *n* = 77 ADVs, *p*s > 0.05). **b** Lag and correlation coefficient of the second peak from the autocorrelation of wrist torque (0.3–1 s after movement onset). The second peaks were detected from trial-based autocorrelations, which were observed within

25–350 ms from lag 0 and whose correlation coefficients were >0.3. According to trial categorization, open black circles indicate that the data were obtained from large ADV trials and closed red circles indicate they were from small ADV trials. Only a few trials (<3.3%) showed second peaks. No significant differences were observed between the large and small ADV trials in mean lag or mean coefficient (two-tailed t-tests, extension, lag, *p* = 0.810, correlation coefficient, *p* = 0.611, df = 242 trials; flexion, lag, *p* = 0.180, correlation coefficient, *p* = 0.769, df = 111 trials). **c** Grand average of the power spectrogram for electromyography (EMG) traces (extensor digitorum-2,3 [ED23] and flexor digitorum superficialis [FDS]) calculated in the same way as in (**a**). No significant difference between trials was found at any frequency (two-tailed paired t-tests with Bonferroni's correction, correction size = 60, *n* = 77 ADVs, *p*s > 0.05). Source data are provided as a Source data file.

(Figs. 2 and 3) could also be set-related activity; the sensori-motor cortex organizes a set signal for the upcoming movement that also prepares to effectively suppress task-irrelevant sensory feedback.

While this reciprocal PSI modulation during the preparation and execution of movements could be functionally relevant, our results also suggest that this global pattern of PAD modulation (Fig. 2) may not always improve motor performance, because there was no reciprocal relationship between the size of PSI and task performance (Fig. 5). As for the antagonistic movement of host muscles, we found that the larger PSI throughout the flexion trials (Fig. 2) was associated with better task performance (Fig. 5). This correspondence suggests that the suppression of task-irrelevant feedback by global PSI facilitation is helpful, i.e. not only for executing antagonistic movements but also for making movements more appropriate for the task requirement by trial-by-trial adjustments.

In contrast, during agonistic movements, we found that the successful trials exhibited larger PSI in the period with dynamic motor output (i.e. AM epoch, Fig. 5). Since PSI suppression was generally found during these movements, this result suggests that global PSI modulation may not help to make proper agonistic movements, in contrast to the case for antagonistic movements.

Here, we suggest that two mechanisms might underly the link between PSI and the task performance of agonistic movements. Firstly, the general suppression of PSI facilitates DR afferent feedback. This general PSI modulation could be a hard-wired mechanism in the neural system for executing agonistic movements, i.e. facilitating proprioceptive input from agonistic muscles, possibly together with the gamma drive. This facilitation may help to shorten muscles to initiate and perform a movement in general, both by increasing the

contribution of relevant spinal reflexes and making prompt proprioceptive feedback to the ascending sensory system for the proper control of the upcoming motor output. However, such hard-wired facilitation may not necessarily assist in agonistic movements depending on the target neural circuit. For example, assuming that the facilitated afferents project to inhibitory interneurons in the spinal reflex circuit, like Ib inhibitory interneurons, then PSI suppression may promote inhibitory autogenic reflex action and thus suppress target muscle activity. Since this additional inhibitory mechanism may interfere with an animal's effort to induce extensor muscle activity during agonistic movements, the secondary PSI facilitation, interpolated on global PSI inhibition, could help to sustain muscle activity for successful movements. This two-step mechanism of PSI modulation further suggests the existence of an independent descending system for general and trial-by-trial PSI modulations, which should be elucidated in future work.

The PSI modulation of muscle afferents reported in this study was different from that seen for cutaneous afferents, which typically emerges as non-specific, stable facilitation regardless of the context of ongoing motor action[17]. The difference in PSI modulation between the two modalities might indicate different roles for proprioceptive and cutaneous feedback in shaping ongoing movements[47]. The proprioceptive signal, compared to the cutaneous signal, may assist or antagonize ongoing motor control, depending on the movement it is involved in or the target neural circuit it projects to. Thus, we propose that PSI is used as a flexible input filter[41] to facilitate task-relevant signals and suppress task-irreverent signals, even for afferent signals from the same origin corresponding to continuously changing behavioural goals. A further advantage of this flexible PSI is its regulation of

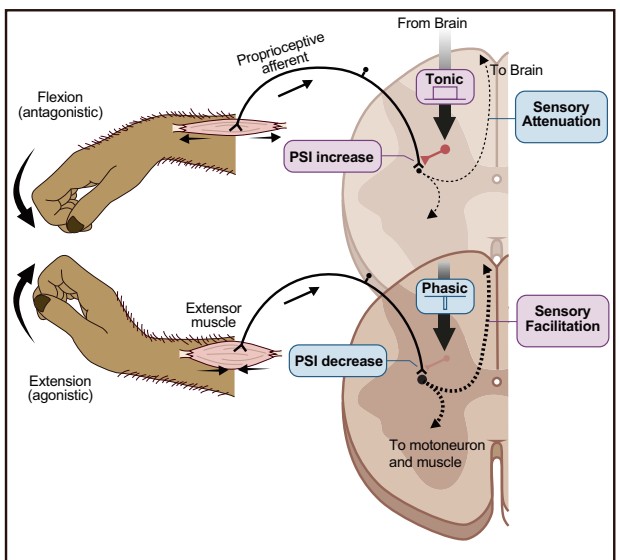

**Fig. 7 | A circuit model for gain modulation of proprioceptive afferent signals by presynaptic inhibition at the spinal cord during agonistic and antagonistic movement.** The gain of the proprioceptive afferent signal to the cervical spinal cord is modulated differentially depending on the role of the host muscles in the context of ongoing movements. For the wrist extensor muscle, wrist flexion is antagonistic, and extension is agonistic. During antagonistic movements (top), descending commands consistently facilitate primary afferent depolarization (PAD) at the afferent terminals, leading to an increase in presynaptic inhibition (PSI). As a result, the afferent-driven activity of first-order spinal interneurons, which project to either the ascending or reflex system, is attenuated, leading to sensory attenuation. In contrast, during agonistic movements (bottom), descending commands suppress PAD, resulting in a decrease in PSI. Consequently, the afferent-driven activity of first-order spinal interneurons is facilitated, leading to sensory facilitation. Created with BioRender.com.

synaptic efficacy without changing the dynamics of postsynaptic circuits. Whether or not a given synaptic input is blocked by PSI, the postsynaptic circuit continues to receive and process synaptic inputs from other sources. In this way, PSI can switch the subject of computation within the same postsynaptic neuronal circuits, so that the circuits function as modules for multiple spatiotemporal and behavioural demands. This modularity is beneficial for supporting an animal's rich behavioural repertoire while using a limited number of neuronal resources.

A recent report[48] showed, in the rodent lumbar spinal cord, that GABA$_A$-generated PAD facilitates spike propagation at the branch point of Ia afferents monosynaptically projecting to motoneurons. At this stage, it is not clear whether this system may also exist in cervical spinal segments controlling forelimb movements, in Ia afferents projecting to spinal interneurons, or the spinal cord of non-human primates and humans. Furthermore, in future studies, it is important to show if this facilitatory mechanism can also dynamically modulate proprioceptive sensory signals, similar to PSI in this paper, or if it secures afferent transmission as a simple homeostatic regulator. Nevertheless, this presynaptic mechanism expands the possibility of the presynaptic regulation of afferent flow during movement. The authors showed the expression of GABA$_A$ receptors in the dorsal spinal cord, but the ventral terminals at motoneurons express GABA$_B$ but not GABA$_A$[48,49]. Thus, proprioceptive afferent input to the spinal cord could be facilitated at the branch points of intraspinal axons[48] and suppressed by PAD (see Supplementary Note 1) by the presynaptic action of GABA$_A$ receptors, and the later suppression is further achieved by PSI generated by the presynaptic GABA$_B$ receptors selectively expressed at the Ia terminals at motoneurons[48,49]. Future intriguing questions are how the excitation-suppression balance of

proprioceptive afferent input is maintained during voluntary movements by harmonizing GABA$_A$-PSI at spinal interneurons versus GABA$_B$-PSI at motoneurons, as well as facilitation at intraspinal axons to motoneurons versus suppression at terminals to interneurons by the GABA$_A$ system.

## Methods

### Animals
Our experiments were approved by the Institutional Animal Care and Use Committees at the National Institute for Physiological Sciences (NIPS), Aichi, Japan and the National Center of Neurology and Psychiatry (NCNP), Tokyo, Japan. Data were obtained from a male and female Macaca fuscata (male: Monkey Y, 7.5 kg at NCNP; female: Monkey O, 5.5 kg at NIPS). During the training and recording sessions, each monkey sat upright in a primate chair with its right arm restrained on an arm rest and its elbow bent at 90°. The monkey's right hand was held in a cast, with its fingers extended and wrist in the mid-supination/pronation position (Fig. 1a). The cast holding the monkey's hand was attached to a servomotor-driven manipulandum that measured flexion-extension torque about the wrist. The left arm was also loosely restrained to the chair[17,18,27,29]. Monkey O was also used in experiments published elsewhere[27].

### Behavioural paradigm
The monkeys performed a wrist flexion-extension task with an instructed delay period using a spring-loaded manipulandum[17,27,50,51] (Fig. 1a). The monkeys were aided by visual feedback of torque at the wrist. The degree to which a magenta rectangular cursor (Fig. 1a, b) deviated from the centre of a display monitor placed in front of them indicated the magnitude of wrist torque that they should exert, with leftward deviation signalling that the monkey should flex. Each trial comprised nine steps: (1) rest, the cursor needed to be inside the centre box for 0.8 s, before the next instruction was delivered. To maintain cursor position, the wrist joint needed to be still without exerting any torque; (2) directional cue, one of the peripheral targets flashed for 0.2 or 0.4 s, signalling the required movement direction; (3) delay, the cursor needed to be inside the centre box for a random period (0.7 ± 0.2 s) for Monkey O, and 0 s for Monkey Y, or the trial was aborted; (4) go cue, the centre box disappeared, signalling the monkey to initiate the movement; (5) active movement, the cursor moved towards the instructed peripheral target (reaction time <0.5 s) in response to the monkey's wrist torque against a spring-like load (5 N·m); (6) active hold, the cursor needed to stay inside the peripheral target for a random period (0.7–1 s, when the torque overshot or undershot the target range without returning to it within 0.1 s, the trial was aborted); (7) end cue, the peripheral target disappeared and the centre box reappeared, signalling permission for return; (8) passive return, the monkey relaxed and the cursor returned towards the centre box as the monkey's wrist torque decreased. The spring servomotor automatically returned the wrist position to the centre unless the monkeys actively fought against it; and (9) reward, after keeping the cursor within the centre box for 0.8 s, a drop of apple sauce was dispensed by a pump (MasterFlex®; Cole-Parmer, Vernon Hills, IL, USA), indicating a successful trial. When the monkey failed any step, the trial was aborted and marked as an error trial. These steps were controlled by special-purposed software (TEMPO; Reflective Computing, Olympia, WA, USA). Behavioural training was performed on an average of 2 h/day, 5 days/week.

### Surgical implants
Following behavioural training, surgery was performed aseptically under 1.5–3.0% sevoflurane anaesthesia with a 2:1 ratio of O$_2$:N$_2$O. To stabilize head and neck movements during the task, head stabilization lugs were cemented to the skull with dental acrylic and anchored to the bone via screws. A resin (Ultem®) recording chamber was

implanted over a hemi-laminectomy in the lower cervical vertebrae (C5–C8). Pairs of stainless-steel wires (AS631; Cooner Wire, Chatsworth, CA, USA) were implanted subcutaneously in 10 or 12 muscles (extensor carpi ulnaris, extensor carpi radialis, extensor digitorum communis, extensor digitorum-2,3, extensor digitorum-4,5, abductor pollicis longus, flexor carpi radialis, flexor digitorum superficialis, palmaris longus, and brachioradialis for both monkeys, and flexor carpi ulnaris and pronator teres only for Monkey Y). Each muscle was identified based on its anatomical location, and was confirmed frequently by checking the finger, wrist, or elbow movements elicited by trains of low-intensity intramuscular stimulation. Sample EMG traces recorded during the task are illustrated in Fig. 1d and Supplementary Fig. 5. Nerve cuff electrodes (Unique Medical Co., Ltd., Tokyo, Japan) were implanted on the radial and median nerves for stimulation and recording ADVs. The inside diameters of the prepared silicone cuffs were 2.5, 3.0, or 4.0 mm. Two or three thin platinum plates were fixed inside the cuff with an inter-electrode distance of 1.5 or 3 mm. A suitably sized nerve cuff (approximately 1.5× the diameter of the nerve[52]) was selected. For the median nerve, a cuff was implanted at 2 cm proximal to the elbow joint. For the radial nerve, two cuffs were implanted in the cutaneous branch (superficial radial nerve) midway between the elbow and wrist: one in the muscle branch (DR nerve) 1.5 cm proximal to the elbow joint and one in the stem (proximal to where the superficial radial and DR nerves branch) 1–2 cm proximal to the elbow joint. Only responses recorded from the DR cuff were used in this study.

### Recording intra-spinal microstimulation-evoked nerve volleys

After recovery from surgery (approximately 10 days), experimental sessions (~5 days/week) for recording ADVs from the DR cuff electrode were conducted by applying repetitive intra-spinal microstimulation (ISMS) while the monkeys performed the task.

Glass-insulated tungsten or Elgiloy microelectrodes (impedance 0.8–1.4 MΩ) were used to record spinal cord surface potentials[53], find appropriate sites, and stimulate the cervical spinal cord (C5–C8, <3 mm from the first observed cell[54]). First, a threshold current for DR stimulation was identified using an incoming volley in the radial nerve cuff electrode while simultaneously measuring spinal cord surface potentials. The microelectrode was then inserted into the spinal cord grey matter to search for ISMS sites[18] that exhibited monosynaptic extracellular responses to DR stimulation (biphasic constant-current pulses, 100 μs/phase, negative-positive at 1 Hz, 1–1.2 times the measured threshold to selectively activate group I afferents[55]). When a response was observed, segmental latency was measured, which was defined as the time from the first spinal cord surface potential peak[56] to the peak time of the peristimulus time histogram of spikes or to the onset of field potential. Any responses with a short segmental latency (<1.5 ms) were deemed putative monosynaptic responses, indicative of a potential site of DR afferent terminals. At these sites, ISMS (3–50 μA, biphasic pulse, 100 μs/phase, 10 Hz) was applied to record ADVs from the DR-cuff electrodes. For Monkey O, a current just above threshold was used, which was approximated by checking the online neurogram average while changing stimulus intensity (3–50 μA). For Monkey Y, a relatively high constant current (20–50 μA) was used that elicited volleys in the nerve.

Data were amplified and filtered by MCP-Plus (Alpha Omega, Nazareth Illit, Israel; high-pass filter = 5 Hz for EMG and 500 Hz for ADV, two-pole Butterworth filter; low-pass filter = 3 kHz for EMG and 10 kHz for ADV, four-pole Butterworth filter) and digitized by DAP4200a/526 (Microstar Laboratories, Crystal Lake, IL, USA) for Monkey O, or amplified, filtered, and digitized by AlphaLabSnR (Alpha Omega, Nazareth Illit, Israel; pre-amplification hardware high-pass filter = 0.5 Hz, two-pole Butterworth filter; pre-amplification hardware low-pass filter = 10 kHz, three-pole Butterworth filter) for Monkey Y. The sampling rates of the nerve-cuff recordings were 40 kHz for

Monkey O or 44 kHz for Monkey Y, those of EMG were 5 kHz for Monkey O or 22 kHz for Monkey Y, and those of wrist torque were 1 kHz for Monkey O or 2.75 kHz for Monkey Y.

### Dissociating antidromic and orthodromic volleys

As the DR nerve is a mixed nerve, i.e. motor and sensory axons are intermingled, any volleys associated with ISMS can be either ADVs (evoked in sensory afferents) or orthodromic volleys (evoked in the axons of motoneurons). To eliminate contamination by orthodromic volleys, stimulus intensity and location were adjusted so that ISMS did not generate any EMG responses. For Monkey O, stimulation current was changed during the experiment and the stimulus-triggered averages of EMG were checked offline for each stimulation current. If significant EMG responses to <10 μA ISMS were found, the data were discarded because such responses indicate that the observed volley is potentially an orthodromic response, as the electrode might be too close to the motoneuron pool[57,58]. For Monkey Y, a fixed stimulus current (>20 μA) was used that was large enough to evoke a volley. Concurrently, the stimulus-triggered averages of EMG were monitored continuously from all recorded muscles, and whenever responses were found in at least one muscle, stimulation was halted and the electrode was moved to identify the next intraspinal site.

### Post hoc definition of behavioural epochs

To define behavioural epochs based on performance, the onset and offset timing for wrist torque and EMG were redefined. Wrist torque was low-pass filtered (<5 Hz), and movement onset was defined as an arbitrary threshold indicated by rapid and steady changes in the derivative of the filtered torque. Movement offset was defined as the first zero-crossing time following the peak torque derivative. EMG signals were rectified, aligned with respect to movement onset, averaged across trials, and low-pass filtered at 10 Hz. The onset of EMG bursts that preceded movement onset was defined as the starting point, with signals exceeding five SDs above the mean EMG amplitude during the Rest epoch for at least 50 ms. For each experimental session, EMG onset for each muscle was computed and representative flexor and extensor muscles (extensor digitorum-2,3 and flexor digitorum superficialis) with the earliest onset latency were selected. EMG offset was calculated by extra-smoothed EMG (low-pass <3 Hz). Local maximum points around movement offset (±300 ms) were extracted and the point that exhibited the maximum difference was defined as EMG offset.

On the basis of these post hoc definitions of behavioural epochs, movement-related epochs were defined as illustrated in Fig. 2a: (1) Rest, the 0.8-s interval before the onset of the cue signal; (2) Delay, from the onset of the cue signal until the onset of EMG activity at a representative agonist muscle; (3) AM, from the onset of EMG activity until the offset of dynamic wrist torque change; (4) AH, from the offset of dynamic movement until the onset of the end cue; and (5) PM, from the onset to offset of passive return torque.

### Classification of successful and error trials

Each trial was judged as perfect if the monkey completed all five behavioural epochs (Figs. 2–4). Moreover, all trials were re-classified according to wrist torque during voluntary movements as successful or three types of error trials: (1) No movement, trials without any detectable changes in torque after the Go signal was delivered. These trials were aborted after the grace period (0.2–0.5 s after the Go signal); (2) Wrong direction, trials in which the movement after the Go signal was in the opposite direction to the instructed direction; and (3) Short hold, trials in which the monkey moved in the correct direction, but did not hold the instructed wrist torque (Fig. 5b–f). Trials were defined as short hold if the minimal torque in the duration from the peak torque time to 1 s after movement onset was smaller than the

arbitrary threshold (which was equivalent to 14.3–37.5% of the torque required to stay within the peripheral target). The trials that included movements in the appropriate direction were deemed as a success if the movements were not classified as short hold. The frequencies of the successful and error trials are summarized in Supplementary Table 2.

### Identification, quantification, and normalization of ADVs

To identify the evoked volleys within the activity of the DR nerve, the overall average of DR responses triggered by all ISMS pulses applied to each intraspinal location was computed. Among them, significant volleys were those that continuously exceeded the baseline (data from −0.3 to −0.1 s relative to ISMS onset) by two SDs for >0.2 ms (Fig. 1), and had onset latencies of <5 ms after ISMS. We found 90 volleys in 113 (Monkey O, 4 months) and 203 (Monkey Y, 10 months) electrode penetrations. After dissociating the orthodromic volleys, 77 volleys were identified exclusively as ADVs. The form of each ADV is summarized in Supplementary Table 1.

ADV sizes were evaluated in terms of their area[17]. From the averaged waveforms, the peak and trough time points from ISMS were defined. Then, the onset and offset of the ADV, and the inflection time between its peak and trough, were defined as the time points when the baseline was crossed (Supplementary Fig. 2a). For monophasic ADVs, the inflection and offset times were identical. All of these predetermined time points were applied to the measurement of individual ISMS responses. ADV area was measured by summing the values of bins from onset to inflection and from inflection to offset. The trough area was numerically inverted because it was negative.

For normalization, we computed the distribution of ADV areas during the Rest epoch as a canonical distribution. After confirming its normal distribution, we used it to transform all other epoch- or event-dependent estimations of ADV areas into $z$-scores. Please note that the mean normalized area during the Rest epoch was represented as zero according to this transformation.

### Quantification of epoch- and event-dependent ADV modulation

To obtain the epoch-dependent modulation of each ADV, each single stimulation in the repetitive ISMS train during a perfect trial was grouped according to the task epochs defined by the post hoc assessment for either flexion or extension. We then compiled the ISMS pulses applied in each behavioural epoch (Fig. 2b) and calculated the area of the response to each ISMS pulse. Finally, those areas were normalized and averaged for each behavioural epoch. This was repeated for all 77 ADVs and their grand average and distribution were obtained (Fig. 2c, $n = 77$) for each behavioural epoch (Fig. 2a). To visualize the bias of their distribution, cumulative summations were calculated (Fig. 2d).

To evaluate event-dependent modulation (Fig. 4, Supplementary Fig. 1c–f), the ISMS pulses were aligned to behavioural events in the perfect trials, i.e., Rest, Delay, AM, AH, or PM onset time, and EMG onset and offset time. A peri-event bin (0.2 s) was then moved in 0.1-s steps from −1 to 1 s relative to the timing of each event, and the ISMS pulse applied within the bin was compiled and used to obtain the normalized and averaged ADV areas ($n = 77$).

Steel's test was applied for multiple comparisons by EZR[59] to compare ADV areas between the Rest epoch and the other behavioural epochs or events. Paired $t$-tests with Bonferroni's correction were used for comparisons between the wrist flexion and extension trials.

### Characterizing the patterns of ADV modulation

To extract the multiple bases for ADV modulation (Fig. 3), ICA was performed using fastICA in the R package. The size of the data matrix was 154 (77 for extension and 77 for flexion) × 4 (epochs), which included the mean of the normalized ADV areas. The maximum number of extractable ICs was four. To obtain the confidence intervals for ICA, we generated a 1000-times bootstrap dataset with replacement from the actual dataset (154 × 4 matrix), and performed ICA. The obtained confidence intervals of ICs and mixing weights were illustrated as dark grey lines (Fig. 3a, b). After performing ICA, the estimated ADV modulations were illustrated based on the extracted ICs and the weight of each IC (mixing weights). For this illustration, we used representative durations for each epoch (Delay, 0.5 s; AM, 0.2 s; AH, 1 s; PM, 0.2 s). The interval between AH termination (end cue) and PM onset was omitted for the illustration. First, square waves were made using the representative epoch durations (100 points/s) and amplitude was defined as the product of the IC and the median of the mixing weights. The square waves were then smoothed using a moving average (10 points/bin = 0.1 s/bin, Fig. 3c).

### Comparison between ADV area and behavioural performance

To reveal the relationship between the area of ADVs ($n = 77$) and task performance, all trials completed from the task start to the AM epoch irrespective of AH success were compiled, i.e. the success and short hold error trials described above. Second, the size of the ADVs evoked in the time period of (i) the Delay period, (ii) 300 ms from EMG onset, and (iii) the AH period was measured. Third, the size of the ADVs evoked within two periods differentially was averaged: (1) dynamic task epoch ([ii], Figs. 5a) and (2) static task epoch ([i + iii], Fig. 5e). To analyse the dynamic task epoch, a 300-ms period was used because it fully covered the AM epoch and it could constantly compile three ADVs in this period, which would provide a more stable calculation of the mean ADV size in this period. To analyse the static task epoch, the trials with an AH epoch were selected; thus, trials with no data in (iii) were excluded from this analysis. These calculations were performed differentially for the flexion and extension trials, and sorted all trials into those with large (largest third of the population: large ADV trials), intermediate (middle third), or small (smallest third: small ADV trials) ADVs. Subsequently, the distributions of several indexes reflecting behavioural performance were compared between the large and small ADV trial groups.

Task performance measurements were as follows: (1) reaction time (latency from cue to movement onset); (2) average torque during the AH epoch; (3) SD of torque during the AH epoch; (4) peak speed (peak value of the torque derivative); (5) peak speed latency (latency of the peak torque derivative from movement onset); (6) peak acceleration (peak value of the second-order derivative of torque); (7) peak acceleration latency (latency of the peak second-order derivative of torque from movement onset); (8) average EMG magnitude during the AH epoch; (9) SD of EMG magnitude during the AH epoch; and (10) success ratio (number of successful trials/total number of trials). Note that reaction time (#1), torque (#2, 3, 4, 6), and EMG (#8 and 9) were normalized for each recording day. Values were computed separately for the 77 ADVs, and the means ± SDs were calculated separately for the three types of trials. Paired $t$-tests were performed to compare the small and large ADV trials. All results obtained in this analysis are shown in Supplementary Fig. 3.

### Evaluation of potential force output oscillations during the task

The frequency spectrograms and auto-correlograms for torque and EMG signals derived from an extensor (extensor digitorum-2,3) and flexor (flexor digitorum superficialis) muscle during the period after movement onset were computed. The assessment window for trial categorization by ADV area was the static task epoch. For this analysis, wrist torque was band-pass filtered (3–100 Hz) and EMG data were band-pass filtered (3–200 Hz) without rectification.

To compute the frequency spectrograms, fast Fourier transformation with Hann smoothing was applied to 1 s of data beginning at movement onset. Single-sided amplitude spectra for fast Fourier

transformation (<60 Hz) were compared between trials with small and large ADVs (Fig. 6a–c). Paired two-tailed t-tests with Bonferroni's correction were performed for all points of the spectrum.

For the auto-correlograms, data from 0.3–1 s after movement onset were extracted and the autocorrelations were calculated trial-by-trial. Autocorrelation results were low-pass filtered (<60 Hz) and the first local maximum with a positive lag (lag ≠ 0), indicating the existence of involuntary oscillations, was detected only if it exhibited a high correlation coefficient (>0.3) and the lag was within 25–350 ms (40–2.86 Hz). Mean coefficients or lags between the small and large ADV trials were compared with t-tests (Fig. 6b).

**Examination of recording stability during nerve cuff recordings**
One monkey (Macaca mulatta, male, 8.5 kg) was additionally trained to perform a wrist flexion-extension task without the Delay and PM epochs. As with the other animals, radial (R) and DR cuffs were implanted, and the stimulus current was applied to the DR nerve and the incoming volleys evoked by each stimulus in the R cuff were recorded while the monkey performed the task (Supplementary Fig. 2a). The data were amplified (×20) and digitized by AlphaLab SNR (Alpha Omega, Nazareth Illit, Israel) at a sampling rate of 22 kHz. Output from the amplifier was band-pass filtered (200–9000 Hz) to monitor the evoked volleys during recording. In the analysis, the volleys in the R nerve were compiled and averaged individually for the behavioural epochs in which they were evoked. Then, the peak-to-peak amplitude of the earliest volleys in each epoch was measured. We specifically compared the normalized values of the amplitudes in the AM and AH epochs for both movement directions with the one in the Rest epoch (zero) by a t-test. Other procedures for the surgical implant, recording, and stimulation were the same as those described above for Monkeys O and Y.

**Probability of responses in postsynaptic neurons of DR afferents**
The modulation of the orthodromic response probability of first-order spinal interneurons to DR afferent was examined by applying a comparable method as in our previous report[27] as shown in Fig. 2e, f. In brief, electrical stimuli were applied to the DR nerve while the monkeys performed the task, and the spiking activity of spinal neurons was recorded. For the neurons exhibiting a response with a latency compatible with the monosynaptic DR input, the spiking activity of each neuron was aligned by stimulation timing and peri-stimulus time histograms (bin = 0.5 ms) were constructed separately for each behavioural epoch and each movement direction. Then, from each peristimulus time histogram, the area of peak response that was significantly greater than baseline (mean firing rate during the 50 ms preceding stimulation [solid horizontal red line in Fig. 2e]) was calculated. Specifically, the area was calculated from the bins between the onset and offset of the peak (filled grey area) that was detected by 2 SDs of the mean (dashed horizontal red line in Fig. 2e).

**Reporting summary**
Further information on research design is available in the Nature Portfolio Reporting Summary linked to this article.

## Data availability
The data are available from the Github repository (https://github.com/saetoma/Tomatsu_NC2023.git). Source data are provided with this paper.

## Code availability
Codes are available at https://github.com/saetoma/Tomatsu_NC2023.git.

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

## Acknowledgements

This work was supported by Grants-in-Aid for Scientific Research from the Ministry of Education, Culture, Sports, Science, and Technology of Japan (18020030, 18500315, 20020029, 23300143, 26120003, 26250013, 19H05724, 19H01092, 23H05488), the Japan Science Technology Agency Precursory Research for Embryonic Science and Technology programme, and commissioned research (no. 22102) from the National Institute of Information and Communications Technology (NICT), Japan (all to K.S.). We thank C. Sasaki, N. Takahashi, K. Takada, and M. Togawa for animal care and technical assistance, M. Kudo for specimen preparation, T. Takei, T. Oya, T. Isa, H. Gomi, and S. Ito for helping with experiments and discussion, and N. Yoshimura and H. Tanaka for consultation on the ICA analysis. We also wish to thank E. Azim for his valuable discussion and comments on the early version of the manuscript.

## Author contributions

K.S. planned the experiments, S.T., G.K., S.K. and K.S. performed the experiments, S.T., G.K. and S.K. analysed the data, and S.T. and K.S. wrote the paper.

## Competing interests

The authors declare no competing interests.
