## [Peer Review File · Nature Communications]

Presynaptic gating of monkey proprioceptive signals for proper motor action.REVIEWER COMMENTS

Reviewer #1 (Remarks to the Author):

In this manuscript, the authors use a surrogate marker of pre-synaptic inhibition (PSI) to study the gating of afferent inputs while monkeys flex and extend their wrists. The goal was to study the gating during active contraction (shortening) of the muscles vs during lengthening, and indeed they show differential effects. In short, the suppression of PSI from extensor muscle afferents during extension can lead to enhanced muscle contraction, and the facilitation of PSI of these afferents during muscle lengthening (flexion) can lead to suppression of unwanted/unnecessary/unhelpful afferent input. Furthermore, a well-thought out analysis is convincing that there's a top-down component to control this particular PSI. The authors propose that PSI acts as a dynamic filter to facilitate movement.

This is a really important study as it sheds light on sensory gating during movement, and the potential importance of such gating in primate movement. The methodology is sound (and previously described by the senior author) – there is no doubt that they are seeing an effect on the antidromic spikes, a marker of PSI.

I only have about 4 comments:

1. I hate to bring this up, but new data from the Bennett lab (which I think is coming out in Nat Neurosci) raises fundamental questions about our understanding of PSI and primary afferent depolarisation. I admit that I haven't yet spent the time necessary to fully understand that paper, but I do wonder how it impacts the interpretations of this manuscript. (This comment applies to the whole manuscript, but some easy fixes are to line 49: "overwhelming evidence;" line 52, maybe "it is thought that GABAergic...")
2. I think that there is too much emphasis in the Results section about the afferents in question being Ia or Ib afferents. We simply do not know this from these nerve cuff electrodes that are recording antidromic volleys in the whole nerve. While an interesting speculation, it seems to me that it belongs in the Discussion as a possible interpretation of the Results.
3. Also for the Discussion, it seems to me that we (?) don't know what the activity is in the particular afferent types during these movements – if this is known, it should be in there – i.e. what activity is being suppressed or not? To me this leads to a whole other line of thought: assuming there is co-activation of alpha and gamma motoneurons (or betas, if you like), then this would occur during extension and the spindles would be "sensitised," leading perhaps to increased Ia (and II) activity – so if there were PSI, it would counter this activity, which seems like a waste. The fact that PSI is suppressed would facilitate the effects of these afferents, and that's really cool. During lengthening, these afferents could be quieter anyway (no gamma tone), but if they fire, their effects are suppressed as this would be

counterproductive. Of course, the authors need not speculate about this in their manuscript, but I thought I'd put it out there!

4. Figures: The bar graphs (e.g. Fig 2b, and 4b and g, but also likely suppl fig 3) need to be replaced. They are uninformative. Firstly, the SEM is inappropriate in most cases in biology, and certainly is here – we are interested in variability in biology, so at minimum show standard deviation. Secondly, bar graphs do not tell the story of variation, even with the standard deviation: better to show box-whisker plots. And thirdly, box-whisker plots, which are informative, can be even more informative if the raw data are plotted alongside or within.

5. Statistics. The experimental unit for monkey studies is necessarily the trials. Experimental sessions were 5 days/week. Was any analysis done on the differences between the data on the different days? Did PSI vary from day to day, or even systematically across the experiment from day 1 to day n? And were the responses in the 2 monkeys generally consistent? I know that these are not easy to answer as statistical power will be low, but there must be a way to gain some insight about these questions?

Minor comments:

1. The summary is a bit unclear in 2 spots: (a) line 35 is a bit strong – the monkeys did not maintain torque in those episodes, not that they could not maintain it (i.e. it's a correlation); (b) line 38, the "only" may not be right – only 1 task was tested.

2. I think the ICA is good – a few words in the Results (p.121) explaining what was done would be helpful. I hope someone with more expertise than I have has a look at this.

3. Line 169: there should be a call-out to Fig 3a, 3b. (By the way, this is a really cool analysis/figure.)

4. Lines 180-182: I'm finding this connection a bit tenuous, and I can't see why it's relevant. If it is relevant, I think further explanation would be helpful.

5. Line 212: this is another example of speculation in the Results that might be better in the Discussion.

6. Line 244: Up to the editors, but the use of the word "first" always rubs me the wrong way – totally unnecessary!

7. Line 396: site

Reviewer #2 (Remarks to the Author):

Presynaptic gating of a monkey proprioceptive signals for proper motor action

The paper examines processing of proprioceptive inputs by the spinal cord in macaques. The authors show how pre-synaptic inhibition of proprioceptive input from extensor muscles is suppressed during a wrist extension movement but largely enhanced during a flexion movement. Further, they show that modulation of pre-synaptic inhibition is under the control by both central and peripheral sources. Lastly, their results highlight how modulation of pre-synaptic inhibition may be required to perform the behaviour successfully. The results help to shed new light on the poorly understood area of spinal circuits and their contribution to sensory processing. However, there are a number of concerns with paper that I highlight below:

Major Concerns

The paper relies solely on interpreting the shapes of anti-dromic spikes elicited from the spinal cord as a proxy for pre-synaptic inhibition. Admittedly, I found it rather difficult to accept the claims of the paper based solely on ADV amplitude as it is unclear to the reader how synaptic depolarization should relate to the shape of the antidromic axonal waveform. The paper only cites one source to support this connection from the 1950s. It is also unclear whether other factors might influence ADV waveforms such as changes in axonal excitability between baseline and movement (i.e. greater excitability of afferent that increases collisions thus distorting the ADV spike), or movement artifacts of the cuff. I would like the paper to include several points: 1) a better connection between ADV waveform and PAD/PSI, 2) provide arguments to rule out other possible causes for the ADV modulation, and 3) explain why a more direct test of pre-synaptic modulation such as DR-evoked responses in spinal cord neurons (such as Seki et al 2003) was not feasible or not pursued.

I find the paper's interpretation surrounding ADVs on error trials to be contradictory. The paper shows that during the extension movement, trials with larger ADVs during the AM and AH epochs result in better trial outcomes than smaller ADVs (Figure 4b, e). This would imply that the ADV reduction observed in Figure 2 for extension movements in the AM epoch may actually be deleterious to behaviour. Yet, the paper seems to state that the observed reduction in ADV amplitude should improve behaviour as the discussion points out (lines 254-256) "... reduced PSI during the agonistic action of host muscles (Fig. 2b and e) reflects the importance of sensory feedback from shortening muscles in initiating dynamic movement". With this logic, trials with smaller ADVs (i.e. more afferent input) should lead to better trial outcomes during the extension movement.

Relatedly, in Figure 4 I am concerned about the analysis of the ADVs during the AH phase for error trials. Specifically, whether the AH epoch on error trials also included time points where the monkey had started to abort the trial and would reflect more of the PM epoch rather than the AH epoch. This is supported by Figure 4d as torque output and muscle activity on error trials is rather transient and seems to reduce during the AH epoch defined in Figure 4f.

I would also appreciate the author's providing more interpretation surrounding the bi-directional modulation of the ADV's in the delay period. It is unclear to me why ADVs should be attenuated differently during the delay period for the flexion and extension movements. It seems unlikely that this is just a version of the active-hold epoch as muscle output appears negligible during this time.

The paper would benefit by providing a richer description of the data as the paper relies heavily on summary bar plots. For example, it would be informative to see the distributions of ADV's presented in Figure 2b as a set of cumulative sums to show how many ADV's actually show a suppression. Likewise it would be useful if the author's included an exemplar unit that was similar to each of the IC components they identified.

An additional analysis the author's might consider concerns whether they can separate out ADVs as Ia and Ib. Specifically, the authors claim IC3 and IC4 resemble the expected profiles for Ia and Ib afferents, respectively. Is there a clear difference in ADV units that are more heavily weighted for IC3 (putative Ia afferents) than IC4, and vice versa? This could be potentially useful for data in Figure 4 as the authors hypothesize a differential role for PAD in Ib and Ia afferents for the extension and flexion movements. I leave it to the author's discretion whether to include this analysis.

Minor Concern

I found that the reporting of the statistics could be improved. First off, it is surprising what quantities are not reported as significant. For example, in Figure 2b only the extension movement in the AM epoch is reported significant but I am surprised the delay and AH epochs for flexion trials are not significant as well. What are their exact p-values? Can the paper also include exact p-values rather than rounded values throughout the manuscript.

Reviewer #3 (Remarks to the Author):

The authors are investigating the functional consequences of presynaptic inhibition of proprioceptive afferent fibers during wrist movements consisting of flexion and extension. They use a well-established methodology to elicit antidromic volleys (ADV's) in the superficial radial (SR) nerve by microstimulation within the spinal cord (see ref. 24). The size of these ADVs provide an indirect measure as to the level of depolarization of the presynaptic terminal (i.e., primary afferent depolarization or PAD), which is caused by presynaptic inhibition. The main difference between this study and that done previously appears to be that the authors consider proprioceptive afferents, rather than cutaneous afferents. The authors first demonstrate differential modulation of PAD during extension and flexion movements. The authors then use independent component analysis in order to reconstruct the temporal modulation of PAD during various tasks. They then demonstrate that PAD modulation is initiated by descending pathways as they occur before actual movement, and that PAD modulation is directly related to movement accuracy.

Overall, it took much effort to understand what this manuscript was about and, while I think that these are novel and interesting results, the quality of the writing does not do justice to them. This is the clearest experimental evidence that presynaptic inhibition is required for behavior, which is very exciting! I've detailed in my major comments below constructive suggestions for improving the manuscript.

The summary and introduction are not well-written. Overall, the authors need to clearly summarize their (esp. ref 24) and other investigators' previous findings, so that the reader can understand what is already known and what novelties this paper brings to the table. Unfortunately, this omission, together with contradictory statements such as "the role of PSI in behaviour remains unclear." (lines 50-51) and "Fink et al.¹³ were the first to report direct evidence that PSI is crucial for the control and stability of voluntary forelimb movements" (lines 59-60), make it difficult to place the results within the bigger picture. In fact, ref. 24 shows that "Sensory input to primate spinal cord is presynaptically inhibited during voluntary movement" as per title.

This manuscript uses similar almost identical methodology than that of a previous publication (ref. 24) not only to measure the primary afferent depolarization (PAD) as mentioned by the authors, but also to the entire behavioral essay including the movements (flexion and extension). The main difference is that the experiments are now focused on proprioceptive inputs as opposed to cutaneous ones. So, why not clearly expose this in the introduction? One could hypothesize that the global enhancement of PAD seen during active movement for cutaneous afferents could be problematic for feedback as to movement details, which would be required for proper execution. However, since proprioceptive afferents have not been investigated to date, they could provide such feedback. The summary should be rewritten as well.

A very interesting result is the analysis of PAD dynamics during correct and failed trials. While this result clearly shows a relationship between PAD and behavior, it is correlational in nature, which does not imply causation. So, the authors need to be careful when they state that "PAD is needed for sustained motor output" (line 185).

The extended figures should become main figures.

Response to Reviewer Comments:

(Author responses in blue, additions to text indicated in *italic*)

Reviewer #1 (Remarks to the Author):

In this manuscript, the authors use a surrogate marker of pre-synaptic inhibition (PSI) to study the gating of afferent inputs while monkeys flex and extend their wrists. The goal was to study the gating during active contraction (shortening) of the muscles vs during lengthening, and indeed they show differential effects. In short, the suppression of PSI from extensor muscle afferents during extension can lead to enhanced muscle contraction, and the facilitation of PSI of these afferents during muscle lengthening (flexion) can lead to suppression of unwanted/unnecessary/unhelpful afferent input. Furthermore, a well-thought out analysis is convincing that there's a top-down component to control this particular PSI. The authors propose that PSI acts as a dynamic filter to facilitate movement.

This is a really important study as it sheds light on sensory gating during movement, and the potential importance of such gating in primate movement. The methodology is sound (and previously described by the senior author) – there is no doubt that they are seeing an effect on the antidromic spikes, a marker of PSI.

I only have about 4 comments:

Comment 1-1:

1. I hate to bring this up, but new data from the Bennett lab (which I think is coming out in Nat Neurosci) raises fundamental questions about our understanding of PSI and primary afferent depolarisation. I admit that I haven't yet spent the time necessary to fully understand that paper, but I do wonder how it impacts the interpretations of this manuscript. (This comment applies to the whole manuscript, but some easy fixes are to line 49: "overwhelming evidence;" line 52, maybe "it is thought that GABAergic...").

Response 1-1:

Thank you very much for the information about their work. We were aware of the study by Hari et al.¹ that was published recently in Nature Neuroscience. Their results show that the Ia primary afferent signal projecting to alpha motoneurons could be modulated at the node of Ranvier, especially around the axonal branch point. Their results were convincing and indeed provide the

interesting possibility of the presynaptic modulation of primary afferent activity using a dual and possibly parallel system in intraspinal axons and their terminals; and by activating GABA_A and GABA_B receptors.

We believe their findings do not contradict the results presented in the present paper as well as our previous work^{2,3,4} at all. First, Hari et al.¹ found GABA_A receptors in the intraspinal axons of Ia afferents and GABA_B at the presynaptic terminals of spinal motoneurons. Therefore, they suggested that PSI at Ia-motoneuron synapses is not mediated by PAD. Most importantly, we explicitly excluded the Ia terminals at these synapses from the excitability testing in this study to prevent the involvement of orthodromic volleys in our analysis (see Methods). Secondly, our previous finding (now also included in Fig. 2e in this paper) about the facilitation of the neural responses evoked by group I afferents was by first-order interneurons, and motoneurons were explicitly excluded from the target of recording⁴. Moreover, our earlier reports about PAD in cutaneous afferents were also focused on their synapses with spinal interneurons, not motoneurons^{2,3,4}. Hari et al.¹, as well as in their earlier report⁵, described the existence of GABA_A receptors in the terminals of Ia afferents located in the more dorsal aspect of the spinal cord (e.g. Fig. 3 in Hari et al.¹).

Rather, we think the novel mechanism found by Hari et al.¹ raises a new possibility and a fascinating question about the presynaptic modulation of afferent input in the spinal cord, e.g. the circuit logic of coordination and functional role of GABA_A-PSI at the dorsal spinal cord versus GABA_B-PSI at the ventral spinal cord, as well as GABA_A-mediated facilitation at the node versus GABA_A-mediated suppression in the terminal. We addressed this new possibility in the final paragraph of the Discussion (lines 374–394):

“A recent report⁴⁶ showed, in the rodent lumbar spinal cord, that GABA_A-generated PAD facilitates spike propagation at the branch point of Ia afferents monosynaptically projecting to motoneurons. At this stage, it is not clear whether this novel system may also exist in cervical spinal segments controlling forelimb movements, in Ia afferents projecting to spinal interneurons, or the spinal cord of non-human primates and humans. Furthermore, in future studies, it is important to show if this novel facilitatory mechanism can also dynamically modulate proprioceptive sensory signals, similar to PSI in this paper, or if it secures afferent transmission as a simple homeostatic regulator. Nevertheless, this novel presynaptic mechanism expands the possibility of the presynaptic regulation of afferent flow during movement. The authors showed the expression of GABA_A receptors in the dorsal spinal cord, but the ventral terminals at motoneurons express GABA_B but not GABA_A^{46, 47}. Thus, proprioceptive afferent input to the spinal cord could be facilitated at the branch points of intraspinal axons⁴⁶ and suppressed by PAD (see Supplementary Note 1) by the presynaptic action of GABA_A receptors, and the later suppression is further achieved by PSI generated by the presynaptic GABA_B receptors selectively expressed at the Ia terminals at motoneurons^{46, 47}. Future intriguing questions

are how the excitation-suppression balance of proprioceptive afferent input is maintained during voluntary movements by harmonizing GABA_A-PSI at spinal interneurons versus GABA_B-PSI at motoneurons, as well as facilitation at intraspinal axons to motoneurons versus suppression at terminals to interneurons by the GABA_A system.”

Comment 1-2:

2. I think that there is too much emphasis in the Results section about the afferents in question being Ia or Ib afferents. We simply do not know this from these nerve cuff electrodes that are recording antidromic volleys in the whole nerve. While an interesting speculation, it seems to me that it belongs in the Discussion as a possible interpretation of the Results.

Response 1-2:

We agree with this comment and have eliminated these speculations from the manuscript.

Comment 1-3:

3. Also for the Discussion, it seems to me that we (I?) don't know what the activity is in the particular afferent types during these movements – if this is known, it should be in there – i.e. what activity is being suppressed or not? To me this leads to a whole other line of thought: assuming there is co-activation of alpha and gamma motoneurons (or betas, if you like), then this would occur during extension and the spindles would be “sensitized,” leading perhaps to increased Ia (and II) activity – so if there were PSI, it would counter this activity, which seems like a waste. The fact that PSI is suppressed would facilitate the effects of these afferents, and that's really cool. During lengthening, these afferents could be quieter anyway (no gamma tone), but if they fire, their effects are suppressed as this would be counterproductive. Of course, the authors need not speculate about this in their manuscript, but I thought I'd put it out there!

Response 1-3:

Thank you very much for this thoughtful suggestion. We do not know the activity profile of type-identified proprioceptive afferents (e.g. Ia or Ib) during this type of movement, and thus, it was difficult to address the precise role of PSI in this paper. However, your idea about linking PSI with the gamma system is intriguing, and it may certainly provide a more systematic understanding of proprioceptive gain modulation. Therefore, we added a section introducing this possibility (lines 283–298):

“Our results showed that descending commands can enhance and suppress the level of PSI directed towards muscle afferents, depending on the role of the host muscles in the context of ongoing movements, i.e. agonist or antagonist. For instance, assuming there is co-activation of

alpha and gamma motoneurons³⁶, voluntary motor commands activate gamma motoneurons to maintain proprioceptive sensitivity during the agonistic action of muscles. The reduced PSI during this agonistic period (Fig. 2b–d) may assist the gamma drive to facilitate further proprioceptive feedback to the spinal cord (Fig. 2e–f), which is crucial for shortening muscles when performing dynamic movements³⁷. Conversely, during the antagonistic action of muscles, the enhanced PSI suggests that feedback from lengthening muscle afferents is less informative or potentially distractive. Possibly, a sensory prediction signal generated by an internal model³⁸ can be more informative than the actual feedback from antagonists, or the gain of muscle spindles could be too low to encode an informative signal because of the presumably lower activity of gamma motoneurons. An intriguing question for future study is how this parallel and somehow redundant operation of proprioceptive gain control^{39, 40} by the gamma and PSI systems is organized in voluntary movements.”

Comment 1-4:

4. Figures: The bar graphs (e.g. Fig 2b, and 4b and g, but also likely suppl fig 3) need to be replaced. They are uninformative. Firstly, the SEM is inappropriate in most cases in biology, and certainly is here – we are interested in variability in biology, so at minimum show standard deviation. Secondly, bar graphs do not tell the story of variation, even with the standard deviation: better to show box-whisker plots. And thirdly, box-whisker plots, which are informative, can be even more informative if the raw data are plotted alongside or within.

Response 1-4:

According to the reviewer’s suggestion, we revised Fig. 2c to show box-whisker plots with raw data. To show the variability more intuitively, we also added density distribution (grey area in Fig. 2c). Furthermore, we replaced the bar plots with box-whisker plots in Fig. 4b, g (now Fig. 5c, g, respectively) and Supplementary Fig. 3.

Comment 1-5:

5. Statistics. The experimental unit for monkey studies is necessarily the trials. Experimental sessions were 5 days/week. Was any analysis done on the differences between the data on the different days? Did PSI vary from day to day, or even systematically across the experiment from day 1 to day n? And were the responses in the 2 monkeys generally consistent? I know that these are not easy to answer as statistical power will be low, but there must be a way to gain some insight about these questions?

Response 1-5:

To test the day-to-day variance of PSI modulation, we performed two-way analysis of variance (epoch \times day) and found no significant interaction (Monkey Y, extension: $p = 0.21$, flexion: $p = 0.96$; Monkey O, extension: $p = 0.87$, flexion: $p = 0.42$). Therefore, we suggest that there is no systematic trend in PSI modulation as a function of recording day. The result is now incorporated into the main text (lines 138–142):

“These principle characteristics of epoch-dependent PAD modulation were consistent in both monkeys (Supplementary Fig. 1a and b), and did not change throughout all recording days (epoch \times day two-way analysis of variance, no significant interaction, for Monkey Y, extension; $p = 0.21$, flexion; $p = 0.96$, for Monkey O, extension; $p = 0.87$, and flexion; $p = 0.42$).”

We addressed consistency between both monkeys in a new figure (Supplementary Figure 1). Although there were minor differences, we found that the main argument made in this paper is consistent in both monkeys. We observed PAD suppression during the AM epoch of the extension trials (Supplementary Figure 1a) and PAD facilitation during the Delay and AH epochs of the flexion trials (Supplementary Figure 1b) in both monkeys. Moreover, we further confirmed that the suppression of PAD during the AM epoch of the extension trials was most dominant when it was aligned with EMG onset (c) than when it was aligned with movement onset (d), EMG offset (e), or return onset (f). We also briefly described this consistency in the text (lines 138–139):

“These principle characteristics of epoch-dependent PAD modulation were consistent in both monkeys (Supplementary Fig. 1a and b)”.

Comment 1-6:

Minor comments:

1. The summary is a bit unclear in 2 spots: (a) line 35 is a bit strong – the monkeys did not maintain torque in those episodes, not that they could not maintain it (i.e. it's a correlation); (b) line 38, the “only” may not be right – only 1 task was tested.

Response 1-6:

We eliminated the text from line 35 in this revision. We agree with point (b) and omitted the word in this revision.

Comment 1-7:

2. I think the ICA is good – a few words in the Results (p.121) explaining what was done would be helpful. I hope someone with more expertise than I have has a look at this.

Response 1-7:

As the reviewer suggested, we added a sentence briefly describing the ICA (lines 159–162):

“To address this point, we extracted the components underlying the observed task-dependent modulation by independent component analysis (ICA), which hypothesized multiple causes to make temporal modulations of PSI at each task epoch and movement (Fig. 2b–d).”

Comment 1-8:

3. Line 169: there should be a call-out to Fig 3a, 3b. (By the way, this is a really cool analysis/figure.)

Response 1-8:

Thank you for this comment. We have added the call-outs (lines 204–206):

“Analysis revealed more prominent transient PAD suppression (as shown in ICI) when the ADVs were aligned to EMG onset (Fig. 4a) than when they were aligned to torque onset (Fig. 4b).”

Comment 1-9:

4. Lines 180-182: I’m finding this connection a bit tenuous, and I can’t see why it’s relevant. If it is relevant, I think further explanation would be helpful.

Response 1-9:

According to this comment and Comment 3-1, the connection between the current results with our previous findings is thoroughly explained in the Introduction (lines 81-90) and Results (lines 145-154, 212-220). In addition, we have added one figure (Fig. 2e, f) introducing an example of an intraspinal site that exhibited the modulation of PAD and neuronal response probability. We believe this revision makes the connection more explicit.

Comment 1-10:

5. Line 212: this is another example of speculation in the Results that might be better in the Discussion.

Response 1-10:

This speculation has now been moved to the Discussion, as suggested.

Comment 1-11:

6. Line 244: Up to the editors, but the use of the word “first” always rubs me the wrong way – totally unnecessary!

Response 1-11:

We eliminated the word.

Comment 1-12:

7. Line 396: site

Response 1-12:

We have corrected this.

Reviewer #2 (Remarks to the Author):

Presynaptic gating of a monkey proprioceptive signals for proper motor action

The paper examines processing of proprioceptive inputs by the spinal cord in macaques. The authors show how pre-synaptic inhibition of proprioceptive input from extensor muscles is suppressed during a wrist extension movement but largely enhanced during a flexion movement. Further, they show that modulation of pre-synaptic inhibition is under the control by both central and peripheral sources. Lastly, their results highlight how modulation of pre-synaptic inhibition may be required to perform the behaviour successfully. The results help to shed new light on the poorly understood area of spinal circuits and their contribution to sensory processing. However, there are a number of concerns with paper that I highlight below:

Comment 2-1:

Major Concerns

The paper relies solely on interpreting the shapes of anti-dromic spikes elicited from the spinal cord as a proxy for pre-synaptic inhibition. Admittedly, I found it rather difficult to accept the claims of the paper based solely on ADV amplitude as it is unclear to the reader how synaptic depolarization should relate to the shape of the antidromic axonal waveform. The paper only cites one source to support this connection from the 1950s. It is also unclear whether other factors might influence ADV waveforms such as changes in axonal excitability between baseline and movement (i.e. greater excitability of afferent that increases collisions thus distorting the ADV spike), or movement artifacts of the cuff. I would like the paper to include several points: 1) a better connection between ADV waveform and PAD/PSI, 2) provide arguments to rule out other possible causes for the ADV modulation, and 3) explain why a more direct test of pre-synaptic modulation such as DR-evoked responses in spinal cord neurons (such as Seki et al 2003) was not feasible or not pursued.

Response 2-1:

To respond to this concern, we added a thorough description of excitability testing using ADVs in whole nerves (Supplementary Note 1) and of the factors affecting ADV waveforms (Supplementary Note 2). Since the reviewer kindly provided three requirements to address this concern, we would like to summarize our response to each point.

Point 1: a better connection between ADV waveforms and PAD/PSI.

Our previous report³ fully discussed the link between ADV waveforms and PAD/PSI. For example, we reported that: (1) the signal recorded by a nerve cuff is independent of the activity of the surrounding muscles (Fig. 2); (2) the volleys induced by intraspinal microstimulation are indeed antidromically conducted (Fig. 4); (3) ADVs could reflect a volley in single (Fig. 5) or multiple primary afferents (Figs. 5 and 6); and (4) ADVs are preferentially recruited in large-diameter, fast conducting afferents (Fig. 7). As for the connection between ADV size and PAD/PSI, we specifically discussed this in a sub-section of the Discussion using the same reference³. The following is a quote from the corresponding section (page 92 in Seki et al. 2009³).

quotation start

What determines the size of the antidromic volley.

As described above, the size of an averaged ADV can be changed during voluntary movement or by applying different strengths of electrical stimuli. Amplitude changes could reflect (1) changes in firing probability of a single SR axon, or (2) variable recruitment of multiple axons with nearly identical CVs, or both.

Changes in the firing probability would pertain if ADVs represent action potentials of single cutaneous afferents, as suggested by the evidence in Figs. 5 and 6. Threshold current to recruit these volleys was usually less than 10 μ A (Figs. 6, 10), which is comparable to the intraspinal threshold for antidromic activation of single muscle afferents by glass microelectrodes placed near the afferent terminals⁶. The areas of ADVs increased in a sigmoid manner with increasing stimulus current (Figs. 5 and 6B,C), probably representing the enhancement of firing probability of these afferent terminals in response to the ISMS. These ADVs saturated at currents (e.g., 5-10 μ A in Figs 6B&C) well below the intensities needed to maximize the area of compound antidromic action potentials of cat lateral gastrocnemius nerve (160 μ A⁷).

Alternatively, ADVs may reflect multiple recruited axons that have essentially the same conduction velocity. In that case, a change in the amplitude of ADV could be ascribed to changes in the number of recruited axons, not only the modulation of firing probability of each terminals. In this study, activation of separate afferents can be detected if their CVs were different, as shown in Fig.6A, consistent with a broad range of CVs (Fig. 7). Peak-to-peak time of a typical ADV is about 0.1-0.3ms. Differences in axon latencies of this duration could significantly reduce the size of averaged ADVs and make them difficult to detect among the noisy background. In the Monkey M,

the ADVs of axons with CVS of 50 and 55m/s would have a latency difference of 0.4ms. Therefore, to create a significant ADV by summation of multiple action potentials, the axons must have essentially identical conduction velocities. It is known that the effective radius of current spread for a 10 μ A stimulus is about 80 μ m for cortical PT cells⁸ or 200-300 μ m for intraspinal axons⁹. Considering the highly divergent pattern of intraspinal branching of primary afferents¹⁰, it is possible that the terminals of SR axons with similar CVs (i.e., differing by less than 5m/s) are localized within this range. Therefore, the modulation of the size of ADV could be ascribed, at least in part, to changes in the recruitment of SR axons with similar CVs.

These two mechanisms are hard to dissociate definitively in our experimental conditions, but fortunately in both cases a change in PAD leads to a co-varying change in the ADV. If the ADV is the action potential of a single afferent, increased PAD would increase its firing probability, leading to increased size of the average ADV. If the ADV represents a superposition of multiple action potentials with the same CV, increased PAD could, in addition, lead to an increase in the number of axons recruited. Since our goal is to detect a changes in PAD, which underlies both mechanisms, we will not attempt to distinguish between them in the rest of the discussion.

quotation end

Although these descriptions were for the superficial radial (SR) nerve, they must be applicable to the DR nerve analysed in the present study. One clear difference is the mixed-nerve nature of the DR nerve (SR is a pure sensory nerve), and the volleys could have been orthodromically and antidromically conducted. Therefore, we discussed the dissociation of ADVs from orthodromic volleys in the *Dissociating antidromic and orthodromic volleys* section in the Methods.

We agree that the connection was not explicit in the previous version, as suggested. Therefore, in the revision, we added the *Measurement of PSI modulation by the size of ADV evoked in muscle afferents* section as a Supplementary Note 1.

Point 2: Provide arguments to rule out other possible causes for the ADV modulation

As discussed in our previous report³ (see above), the modulation of ADV size could represent two factors: changes in the firing probability of a single DR axon and/or variable recruitment of multiple DR axons with nearly identical conduction velocities. These two physiological factors are indistinguishable, and both were certainly reflected in the results of the current study. On the basis of this assumption, we listed our arguments to rule out the potential involvement of four other factors on ADV modulation, i.e. a) cross-talk from the activity of muscles surrounding the DR nerve cuffs, b) contamination from orthodromic volleys in DR motor axons, c) change of recording efficacy of the nerve cuffs due to wrist movement, and d) distorting effect of ADV waveforms by collision with orthodromic spikes in the same axon. See Supplementary Note 2 for the full text of these arguments and Supplementary figure 2 for the result of additional

experiments to address this comment. We also described these points in lines 142-144 and 661-675.

Point 3: Explain why a more direct test of pre-synaptic modulation such as DR-evoked responses in spinal cord neurons (such as Seki et al 2003) was not feasible or not pursued.

The rationale for why we used ADVs to investigate the presynaptic modulation of DR afferents is now described extensively in the Introduction (see also Response 3-1, 3-2).

As suggested here, in the experiments described in our previous report², we have mainly reported the decreased probability of the monosynaptic responses of spinal interneurons to cutaneous (SR) stimuli during movement. The probability of monosynaptic responses could be influenced by the presynaptic modulation and base excitability of spinal interneurons. Since we found the suppression of responses when the excitability of a cell was highest in the task, we considered that the main cause should be PSI. To support this conclusion, we fully described the results of excitability testing on SR afferent terminals in the subsequent report³.

As for the response of first-order spinal interneurons to DR stimulation, we have reported contrasting results with what has been found for SR responses⁴. We found the facilitation of DR responses during wrist extension, not suppression. In this case, response facilitation is observed when the base excitability of a cell is highest in the task. Thus, we have no evidence showing if the modulation preferentially reflects a presynaptic, not postsynaptic, mechanism; response facilitation might simply represent the increased intrinsic excitability of those neurons.

Therefore, in this manuscript, rather than repeating the experiment reported previously⁴, we focused on excitability testing to examine whether the observed modulation of the DR response was generated by the presynaptic mechanism, i.e. reduced PSI. Obviously, it would be desirable if we could examine DR input modulation and excitability testing for every single intraspinal site in this paper. However, the available experimental techniques do not allow us to maintain a constant condition when recording the DR response of interneurons as well as stimulating the same DR afferent terminals for a sufficient duration. Therefore, we believe it was reasonable to perform the two experiments separately. Anyhow, in the data pool of this paper, we found eight first-order spinal neurons that were recorded at the same electrode position as for ADV recording during at least a few extension trials. Five of them indicated an increased response probability in the AM epoch in extension movements, but they were too small in number to perform statistical analysis. One of the five is newly illustrated in Fig. 2e and f, and the analysis procedure is described in this revision (lines 677-690). This example indeed indicates an increased response probability to DR stimulation when the ADVs are mostly decreased (black dots in Fig. 2c). As a result, in this paper, we can conclude that the decreased ADVs represented reduced PSI, and suggest that the facilitation of DR response observed in the previous paper was also produced by reduced PSI. In addition, we found the facilitation of ADVs during the flexion trials suggesting increased PSI at the terminals of the

group I afferents of antagonistic muscles. The latter conclusion cannot be made by the findings of our previous reports⁴; although we found a lower probability of DR responses during flexion⁴, it could have been generated simply by lower intrinsic excitability, and not by enhanced PSI.

Comment 2-2:

I find the paper's interpretation surrounding ADVs on error trials to be contradictory. The paper shows that during the extension movement, trials with larger ADVs during the AM and AH epochs result in better trial outcomes than smaller ADVs (Figure 4b, e). This would imply that the ADV reduction observed in Figure 2 for extension movements in the AM epoch may actually be deleterious to behaviour. Yet, the paper seems to state that the observed reduction in ADV amplitude should improve behaviour as the discussion points out (lines 254-256) "... reduced PSI during the agonistic action of host muscles (Fig. 2b and e) reflects the importance of sensory feedback from shortening muscles in initiating dynamic movement". With this logic, trials with smaller ADVs (i.e. more afferent input) should lead to better trial outcomes during the extension movement.

Response 2-2:

We appreciate this careful comment. Indeed, we agree that the link between PAD suppression during extension movements and its behavioural outcome was contradictory. In the revision, we resolved this contradiction by assuming two-step PAD modulation during extension. We added a new section to the Discussion to reflect this argument (lines 333–355):

"In contrast, during agonistic movements, we found that successful trials were preceded by larger PSI (Fig. 5). Since PSI suppression was generally found during these movements, this result suggests that global PSI modulation may not help to make proper agonistic movements, in contrast to the case for antagonistic movements.

Here, we suggest that two mechanisms might underly the link between PSI and the task performance of agonistic movements. Firstly, the general suppression of PSI facilitates DR afferent feedback. This general PSI modulation could be a hard-wired mechanism in the neural system for executing agonistic movements, i.e. facilitating proprioceptive input from agonistic muscles, possibly together with the gamma drive. This facilitation may help to shorten muscles to initiate and perform a movement in general, both by increasing the contribution of relevant spinal reflexes and making prompt proprioceptive feedback to the ascending sensory system for the proper control of the upcoming motor output. However, such hard-wired facilitation may not necessarily assist in agonistic movements depending on the target neural circuit. For example, assuming that the facilitated afferents project to inhibitory interneurons in the spinal reflex circuit, like Ib inhibitory interneurons, then PSI suppression may promote inhibitory autogenic reflex action and thus suppress target muscle activity. Since this additional inhibitory mechanism may interfere with an animal's effort to induce extensor

muscle activity during agonistic movements, the secondary PSI facilitation, interpolated on global PSI inhibition, could help to sustain muscle activity for successful movements. This two-step mechanism of PSI modulation further suggests the existence of an independent descending system for general and trial-by-trial PSI modulations, which should be elucidated in future work.”

Comment 2-3:

Relatedly, in Figure 4 I am concerned about the analysis of the ADVs during the AH phase for error trials. Specifically, whether the AH epoch on error trials also included time points where the monkey had started to abort the trial and would reflect more of the PM epoch rather than the AH epoch. This is supported by Figure 4d as torque output and muscle activity on error trials is rather transient and seems to reduce during the AH epoch defined in Figure 4f.

Response 2-3:

We apologize for this confusion. In the previous version, our description of the analysis of the error trials in the flexion condition (Fig. 4f–j in the original version, Fig. 5e–h in this revision) was insufficient. In fact, we used the trials that included the Delay and AH epochs in this analysis. Therefore, the number of error trials used for this analysis was highly limited, and they had relatively longer hold periods than those that appeared in the grand average of the short hold error trials. This means we only included ‘almost’ successful trials. Therefore, the AH epoch in this analysis never included any obvious PM epoch. As for Fig. 4i in the original version, we noticed that we selected inappropriate trials. Therefore, we replaced them with the trials we actually used for the analysis. To avoid confusion, we also revised the description of the classification of the successful and error trials in the Methods (lines 538–551):

“Each trial was judged as perfect if the monkey completed all five behavioural epochs (Figs. 2–4). Moreover, all trials were re-classified according to wrist torque during voluntary movements as successful or three types of error trials: (1) No movement, trials without any detectable changes in torque after the Go signal was delivered. These trials were aborted after the grace period (0.2–0.5 s after the Go signal); (2) Wrong direction, trials in which the movement after the Go signal was in the opposite direction to the instructed direction; and (3) Short hold, trials in which the monkey moved in the correct direction, but did not hold the instructed wrist torque (Fig. 5b and 5f). Trials were defined as short hold if the minimal torque in the duration from the peak torque time to 1 s after movement onset was smaller than the arbitrary threshold (which was equivalent to 14.3–37.5% of the torque required to stay within the peripheral target). The trials that included movements in the appropriate direction were deemed as a success if the movements were not classified as short hold. The frequencies of the successful and error trials are summarized in Supplementary Table 2.”

We also revised the description of the procedure of the analysis (lines 611–624):

“To reveal the relationship between the area of ADVs ($n = 77$) and task performance, all trials completed from the task start to the AM epoch irrespective of AH success were compiled, i.e. the success and short hold error trials described above. Second, the size of the ADVs evoked in the time period of (i) the Delay period, (ii) 300 ms from EMG onset, and (iii) the AH period was measured. Third, the size of the ADVs evoked within two periods differentially was averaged: (1) dynamic task epoch ([ii], Fig. 5a) and (2) static task epoch ([i + iii], Fig. 5e). To analyse the dynamic task epoch, a 300-ms period was used because it fully covered the AM epoch and it could constantly compile three ADVs in this period, which would provide a more stable calculation of the mean ADV size in this period. To analyse the static task epoch, the trials with an AH epoch were selected; thus, trials with no data in (iii) were excluded from this analysis. These calculations were performed differentially for the flexion and extension trials, and sorted all trials into those with large (largest third of the population: large ADV trials), intermediate (middle third), or small (smallest third: small ADV trials) ADVs.”

Comment 2-4:

I would also appreciate the author’s providing more interpretation surrounding the bi-directional modulation of the ADV’s in the delay period. It is unclear to me why ADVs should be attenuated differently during the delay period for the flexion and extension movements. It seems unlikely that this is just a version of the active-hold epoch as muscle output appears negligible during this time.

Response 2-4:

As suggested, the size of ADVs tended to decrease during the extension trials and increase during the flexion trials, both in the delay period (Fig. 2c, Supplementary Figure 1a and b). This observation was also supported by the results of ICA (e.g. IC2, Fig. 3). In the revision, we added a paragraph to the Discussion to address our suggestion of ADV modulation in the delay period (lines 299–322):

“This reciprocal modulation of PSI in the flexion and extension tasks was already observed in the Delay period, i.e. a period for preparing future movements (Figs. 2 and 3). Since no overt motor action has been initiated yet in this period, the relevance of PSI modulation should be different from that during the movements discussed above. It is widely known that neurons in the primate sensorimotor cortex^{41, 42, 43} show preparatory activity for upcoming movements. It is likely that the corticospinal input to these preparatory signals could be the source of PSI modulation. Then, why does the cortical motor preparatory signal modulate PSI in a reciprocal way, similar to the one found during movement? We suggest that the suppression of PSI before agonistic extensor movements (Fig. 2) aides proper movement initiation. In an earlier report in human subjects using a reflex-testing battery, Hultborn et al.⁴⁴ suggested that the Ia input from contracting muscles is facilitated by

decreased PSI at movement onset. Since this facilitation is reported at the very onset of EMG activity, we assume that the descending command suppressed PSI even before the moment of movement initiation as a set or priming signal for an upcoming movement. If this is the case, the gain of proprioceptive input in this preparatory period should be set at below, but close to, the recruitment threshold of agonistic motoneurons by PSI suppression, probably together with the modulation of the excitability of gamma motoneurons. Supporting this suggestion, we found the facilitation of DR-evoked monosynaptic responses during the Delay period (Fig. 2e-f), which is consistent with our previous report²⁷. Conversely, the potential role of PSI facilitation observed before antagonistic flexor movements (Fig. 2) could also be set-related activity; the sensori-motor cortex organizes a set signal for the upcoming movement that also prepares to effectively suppress task-irrelevant sensory feedback.”

Comment 2-5:

The paper would benefit by providing a richer description of the data as the paper relies heavily on summary bar plots. For example, it would be informative to see the distributions of ADV's presented in Figure 2b as a set of cumulative sums to show how many ADV's actually show a suppression. Likewise it would be useful if the author's included an exemplar unit that was similar to each of the IC components they identified.

Response 2-5:

According to this suggestion, we made cumulative sum plots in Fig. 2d (lines 584-585). We appreciate this suggestion because the new plot is highly informative, and it clearly demonstrates that the negative values were dominant in the extension condition, especially in the AM epoch. We also changed the bar plots in the previous version into box-whisker plots with raw data and density distribution (Fig. 2c), and with box-whisker plots in Fig. 5c, g and Supplementary Fig. 3 (see also Response 1-4). We also added an exemplar ADV unit next to each IC component in Fig. 3d, according to this suggestion.

Comment 2-6:

An additional analysis the author's might consider concerns whether they can separate out ADVs as Ia and Ib. Specifically, the authors claim IC3 and IC4 resemble the expected profiles for Ia and Ib afferents, respectively. Is there a clear difference in ADV units that are more heavily weighted for IC3 (putative Ia afferents) than IC4, and vice versa? This could be potentially useful for data in Figure 4 as the authors hypothesize a differential role for PAD in Ib and Ia afferents for the extension and flexion movements. I leave it to the author's discretion whether to include this analysis.

Response 2-6:

Thank you for this constructive suggestion. However, since we decided to eliminate our speculative discussion about Ia and Ib afferents from this manuscript in response to Comment 1-2, we think adding this result makes little sense for the main argument. We would like to reserve the suggested analysis for future publication.

Comment 2-7:

Minor Concern

I found that the reporting of the statistics could be improved. First off, it is surprising what quantities are not reported as significant. For example, in Figure 2b only the extension movement in the AM epoch is reported significant but I am surprised the delay and AH epochs for flexion trials are not significant as well. What are their exact p-values? Can the paper also include exact p-values rather than rounded values throughout the manuscript.

Response 2-7:

According to the reviewer's comment, we added exact p-values to the text for all statistical analyses. For example, in the Results (lines 123–137):

“For example, in the wrist extension trials, PAD suppression occurred during the Active Movement (AM) epoch (Fig. 2c, $t = 3.06$, $p = 0.012$, paired two-tailed t -test with Bonferroni's correction, compared with Rest), suggesting that afferent input from extensor muscles was facilitated during dynamic wrist extension. In contrast, in the flexion trials, PAD facilitation occurred during static task epochs (i.e. the Delay and Active Hold [AH] epochs). Mean PAD was positive in the Delay and AH epochs; PAD facilitation was significant when the ADVs in both epochs were aggregated ($t = 2.46$, $p = 0.015$, paired two-tailed t -test), although PAD facilitation did not reach significant levels when they were compared individually ($ps = 0.53$ and 0.22 , respectively, paired two-tailed t -test with Bonferroni's correction, compared with Rest). Furthermore, the global average of PAD was larger in the flexion trials than in the extension trials (§ in Fig. 2c, $t = 2.51$, $p = 0.012$, paired two-tailed t -test, compared between the aggregated data of the flexion and extension trials). As the Delay and AH epochs comprised 75–87.5% of the movement-related periods, these results suggest that PAD during the static task epochs is larger in the flexion trials than in the extension trials.”

We also made similar changes in lines 141, 142, 239, 244, 247, 249, 262-268.

Reviewer #3 (Remarks to the Author):

The authors are investigating the functional consequences of presynaptic inhibition of proprioceptive afferent fibers during wrist movements consisting of flexion and extension. They use a well-established methodology to elicit antidromic volleys (ADV) in the superficial radial (SR) nerve by

microstimulation within the spinal cord (see ref. 24). The size of these ADVs provide an indirect measure as to the level of depolarization of the presynaptic terminal (i.e., primary afferent depolarization or PAD), which is caused by presynaptic inhibition. The main difference between this study and that done previously appears to be that the authors consider proprioceptive afferents, rather than cutaneous afferents. The authors first demonstrate differential modulation of PAD during extension and flexion movements. The authors then use independent component analysis in order to reconstruct the temporal modulation of PAD during various tasks. They then demonstrate that PAD modulation is initiated by descending pathways as they occur before actual movement, and that PAD modulation is directly related to movement accuracy.

Overall, it took much effort to understand what this manuscript was about and, while I think that these are novel and interesting results, the quality of the writing does not do justice to them. This is the clearest experimental evidence that presynaptic inhibition is required for behavior, which is very exciting! I've detailed in my major comments below constructive suggestions for improving the manuscript.

Comment 3-1:

The summary and introduction are not well-written. Overall, the authors need to clearly summarize their (esp. ref 24) and other investigators' previous findings, so that the reader can understand what is already known and what novelties this paper brings to the table. Unfortunately, this omission, together with contradictory statements such as "the role of PSI in behaviour remains unclear." (lines 50-51) and "Fink et al.13 were the first to report direct evidence that PSI is crucial for the control and stability of voluntary forelimb movements" (lines 59-60), make it difficult to place the results within the bigger picture. In fact, ref. 24 shows that "Sensory input to primate spinal cord is presynaptically inhibited during voluntary movement" as per title.

Response 3-1:

We agree with this comment that the previous version insufficiently introduced previous works by ourselves and other groups, and thus, it was misleading. Accordingly, we entirely revised the Abstract (lines 17–25) and Introduction (lines 30–46 and 56-102) to introduce the link between previous work and the objective of this study.

Comment 3-2:

This manuscript uses similar almost identical methodology than that of a previous publication (ref. 24) not only to measure the primary afferent depolarization (PAD) as mentioned by the authors, but also to the entire behavioral essay including the movements (flexion and extension). The main

difference is that the experiments are now focused on proprioceptive inputs as opposed to cutaneous ones. So, why not clearly expose this in the introduction? One could hypothesize that the global enhancement of PAD seen during active movement for cutaneous afferents could be problematic for feedback as to movement details, which would be required for proper execution. However, since proprioceptive afferents have not been investigated to date, they could provide such feedback. The summary should be rewritten as well.

Response 3-2:

We totally agree with this comment. In this revision, we entirely rewrote the Abstract, Introduction, and Discussion to reflect this comment (see lines 19–20, 56-102, 356-362 for example).

Comment 3-3:

A very interesting result is the analysis of PAD dynamics during correct and failed trials. While this result clearly shows a relationship between PAD and behavior, it is correlational in nature, which does not imply causation. So, the authors need to be careful when they state that “PAD is needed for sustained motor output” (line 185).

Response 3-3:

We changed the section heading to *PAD associated with sustained motor output* (line 222). In addition, we carefully rewrote the text in the revision to reflect this concern (lines 223-253).

Comment 3-4:

The extended figures should become the main figures.

Response 3-4:

Extended Figures 2 and 4 are now included in the main manuscript (Figs. 2b and 6, respectively). We decided that some extended figures (Extended Figs. 1 and 3) should be kept in the Supplemental Figures (Supplemental Figures 4 and 3 in this revision, respectively) because they are supplemental to the main argument of this paper.

References

1. Hari K, *et al.* GABA facilitates spike propagation through branch points of sensory axons in the spinal cord. *Nat Neurosci*, (2022).
2. Seki K, Perlmutter SI, Fetz EE. Sensory input to primate spinal cord is presynaptically inhibited during voluntary movement. *Nat Neurosci* **6**, 1309-1316 (2003).

3. Seki K, Perlmutter SI, Fetz EE. Task-dependent modulation of primary afferent depolarization in cervical spinal cord of monkeys performing an instructed delay task. *J Neurophysiol* **102**, 85-99 (2009).
4. Confais J, Kim G, Tomatsu S, Takei T, Seki K. Nerve-Specific Input Modulation to Spinal Neurons during a Motor Task in the Monkey. *J Neurosci* **37**, 2612-2626 (2017).
5. Lucas-Osma AM, *et al.* Extrasynaptic alpha5GABAA receptors on proprioceptive afferents produce a tonic depolarization that modulates sodium channel function in the rat spinal cord. *J Neurophysiol* **120**, 2953-2974 (2018).
6. Duenas SH, Rudomin P. Excitability changes of ankle extensor group Ia and Ib fibers during fictive locomotion in the cat. *Exp Brain Res* **70**, 15-25 (1988).
7. Enriquez-Denton M, Manjarrez E, Rudomin P. Persistence of PAD and presynaptic inhibition of muscle spindle afferents after peripheral nerve crush. *Brain Res* **1027**, 179-187 (2004).
8. Stoney SD, Jr., Thompson WD, Asanuma H. Excitation of pyramidal tract cells by intracortical microstimulation: effective extent of stimulating current. *J Neurophysiol* **31**, 659-669 (1968).
9. Gustafsson B, Jankowska E. Direct and indirect activation of nerve cells by electrical pulses applied extracellularly. *J Physiol* **258**, 33-61 (1976).
10. Willis WD, Coggeshall RE. *Sensory mechanism of the spinal cord*, second edn. Plenum Press (1991).
11. Hultborn H, Meunier S, Pierrot-Deseilligny E, Shindo M. Changes in presynaptic inhibition of Ia fibres at the onset of voluntary contraction in man. *J Physiol* **389**, 757-772. (1987).

REVIEWER COMMENTS

Reviewer #1 (Remarks to the Author):

This is a manuscript that I previously reviewed.

The authors have done an incredible, thorough job in responding to the previous comments and revising their work. In particular, my comment 1-3 (which, incidentally, I could barely follow in re-reading it now): their new paragraph is really interesting.

My only concern is Figure 2c (but I like the box-whisker plot now). I'm having some trouble understanding the statistical approach here. I think it's the term "global average" by which I think they are referring to the averages of the aggregated data. If this is what they've done, it may or may not be useful but worth reporting (does it make sense to aggregate the phases?), but then maybe they should show the aggregated data as well? Also, what is the rationale for combining Delay and AH phases together in the analysis?

And one semantic comment: using the phrase that a test did not "reach" statistical significance (twice in this manuscript, including Fig 2c) implies that the authors think that there is a difference between the two groups of data, but that, perhaps, there's not enough power (or some other explanation). But that is not what the data are objectively saying: the data are telling us that we do not have the statistical grounds to reject the null hypothesis that the two groups are the same.

Reviewer #2 (Remarks to the Author):

Presynaptic gating of monkey proprioceptive signals for proper motor action

In this manuscript, the authors explore how proprioceptive feedback may be modulated at first-order spinal interneurons by pre-synaptic inhibition. The authors have done an excellent job of addressing my previous concerns with the manuscript. However, there are a few additional concerns I would like the authors to address.

A major finding of the present study was that PAD modulation emerged prior to a motor action suggesting a contribution from a descending cortical pathway. Although I don't disagree that this pathway plays a key role, I find the evidence supporting this claim is rather weak. One piece of evidence is that ICA analysis revealed components (IC2) that exhibited PAD modulation in the delay period that was direction specific. However, it's unclear whether these components are genuinely meaningful or whether they are simply the dimensionality technique trying to find structure in the noise. This result would be better bolstered if the manuscript included a statistical analysis to demonstrate that the results are not spurious. At the very least the ICs could include confidence intervals by bootstrapping the means of each ADV across trials. This method can also provide a probability value by asking how many times did the bootstrap distributions for the extension and flexion movements show the expected result (i.e. for IC2 flexion > 0 and extension trials <0). However, I leave it to the authors to determine the best method of statistically verifying their ICA results.

In a similar vein, the manuscript's interpretation of Figure 5 is a bit incorrect. In particular, Figure 5e-g shows that PAD modulation can correlate with trial success during the static hold epoch. However, the authors interpret Figure 5 in the discussion as (line 333) "In contrast, during agonistic movements, we found that successful trials were preceded by larger PSI (Fig. 5)." (see also line 251 in the Results). This interpretation is rather problematic as the data actually reflect the pooling of the Delay and Active Hold epochs (line 228, i.e. static hold epoch). As such, it is unclear to the reader whether this correlation truly reflects modulation in the Delay epoch or might be largely generated by the Active Hold epoch. The manuscript would benefit from not pooling the Delay and Active Hold periods and showing the results for each epoch separately.

The manuscript should acknowledge the apparent contradiction presented throughout the paper about whether ADV modulation is present in the delay period. The results presented in Figure 2C suggest no statistically significant modulation in the Delay period for either the extension or flexion movements (though a small positive bias exists for the flexion movement). Similarly, Figure 4A indicates significant ADV modulation starting 100ms after EMG onset which is after the Delay period. In contrast, results in Figure 3 and 5 tell a different story with significant modulation in the Delay period. I recognize that Figure 2C and Figure 4A reflect trial and ADV averages that likely remove finer modulation effects that are more readily observed using ICA analysis. It would be a useful point to acknowledge this point and may be a useful additional motivation in the Results for the ICA analysis.

Minor points

Caption for Figure 5 requires significantly more detail. Also, the layout of Figure 5 causes problems. I realize now the red-blue color scheme denotes flexion and extension movements however, with the way the figure is laid out I had initially assumed it corresponded to the AM epoch and Static Hold epoch based on sub panels A and E. This initially confused me when trying to interpret sub panels D and H. Thus, The figure should present the data from the same direction in A and E to prevent confusion.

Currently, the manuscript only describes that ADV shape correlates with the amount of PAD. Can the authors be more specific by including at the beginning of the Results that the size of the ADV shape positively correlates with the amount of PAD.

In the Results, reporting of t-tests should also include an explicit statement of the degrees of freedom and the size of the Bonferroni correction factor that was used.

Can the author's include more details about the kinematic and EMG filters (i.e. type and poles, Lines 516 and 521).

Reviewer #3 (Remarks to the Author):

The authors have satisfactorily addressed my comments.

Response to Reviewer Comment:

(Author responses in blue, addition to text indicated in *italic*)

Reviewer #1 (Remarks to the Author):

This is a manuscript that I previously reviewed.

The authors have done an incredible, thorough job in responding to the previous comments and revising their work. In particular, my comment 1-3 (which, incidentally, I could barely follow in re-reading it now): their new paragraph is really interesting.

Comments 1-1.

My only concern is Figure 2c (but I like the box-whisker plot now). I'm having some trouble understanding the statistical approach here. I think it's the term "global average" by which I think they are referring to the averages of the aggregated data. If this is what they've done, it may or may not be useful but worth reporting (does it make sense to aggregate the phases?), but then maybe they should show the aggregated data as well? Also, what is the rationale for combining Delay and AH phases together in the analysis?

Answer 1-1.

Thank you for your helpful comments. In the revised manuscript, we avoided the use of the term "global average" because we used aggregated data to describe the results. We have presented the average values of the four epochs in the revised manuscript (lines 144–147, red font indicates new text):

"We found that PAD in the flexion trials was larger than in the extension trials (§ in Fig. 2c, mean = -0.13 and 0.19, respectively, df = 306, t = 2.51, p = 0.012, paired two-tailed t-test, compared between the aggregated data of the flexion and extension trials)..."

We also presented the average values of the Delay and AH epochs in the flexion and extension trials (lines 157–163):

"We found that the aggregated PADs of the Delay and AH epochs was significantly larger than in the Rest period in the flexion trials (mean = 0.29, df = 153, t = 2.46, uncorrected p = 0.02, paired two-tailed t-test with Bonferroni's correction, correction size = 2). We did not observe a comparable modulation in the extension trials (mean = -0.10, df = 153, t = 0.84, uncorrected p = 0.40, paired two-tailed t-test with Bonferroni's correction, correction size = 2, compared with Rest),..."

With respect to the rationale for combining the Delay and AH phases, we wanted to use them as a representative epoch that characterizes the different modulation of PAD between the flexion and extension trials.

We excluded the dynamic epochs from the evaluation of directional PAD modulation because we had already identified non-directional modulation (lines 123–142) in this epoch, and it was likely that it would introduce a bias in the directional comparisons.

“First, we found the distinct modulation of PAD (Fig. 2b–d) depending on the behavioural epoch (Fig. 2a). In the wrist extension trials, PAD suppression occurred during the Active Movement (AM) epoch (Fig. 2c, mean = -0.44, df = 76, t = 3.06, uncorrected p = 0.003, paired two-tailed t-test with Bonferroni’s correction, correction size = 4, compared with Rest), suggesting that afferent input from extensor muscles is facilitated during dynamic wrist extension. The other epochs did not differ from the Rest period (Delay, mean = -0.12, t = 0.81, p = 0.42; AH, mean = -0.07, t = 0.41, p = 0.68; PM, mean = 0.11, t = 0.79, p = 0.43, df = 76, p-values are uncorrected, paired two-tailed t-test with Bonferroni’s correction, correction size = 4). In the wrist flexion trials, it was noteworthy that the sole negative mean was observed in the AM epoch, in common with the wrist extension trials, although none of the epochs differed from the Rest period (Delay, mean = 0.26, t = 1.44, p = 0.13; AM, mean = -0.03, t = 0.21, p = 0.83; AH, mean = 0.33, t = 1.95, p = 0.05; PM, mean = 0.04, t = 0.26, p = 0.79, df = 76, p-values are uncorrected, paired two-tailed t-test with Bonferroni’s correction, correction size = 4). The specificity of the AM epoch was also detected in the cumulative summation plot of PAD size (Fig. 2d, blue and red solid lines); They were clearly separate from the other traces of respective movement directions and biased to negative values. These results led us to conclude that PSI of the muscle nerve input during voluntary movements has a non-directional characteristic, i.e. it facilitates proprioceptive input during dynamic movements.”

Indeed, the grand average of all epochs for both movement directions did not differ from that of the Rest period (extension, mean = -0.13, df = 307, t = 1.66, uncorrected p = 0.10; flexion, mean = 0.19, df = 307, t = 1.88, uncorrected p = 0.06, paired two-tailed t-test with Bonferroni’s correction, correction size = 2), while we found a significant difference between the flexion and extension trials (lines 144–147), indicating their average came out even probably due to the different polarity of epoch-dependent modulation within the task. Therefore, the grand average of all epochs would not be appropriate for evaluating movement direction-dependent modulation.

In the revised manuscript, we described this point in the text (lines 143–157, Please note that the reference number after #31 were shifted because we added two references):

“Next, we compared PAD modulation between the different movement directions. We found that PAD in the flexion trials was larger than in the extension trials (§ in Fig. 2c, mean = -0.13 and 0.19, respectively, df = 306, t = 2.51, p = 0.012, paired two-tailed t-test, compared between the aggregated data of the flexion and extension trials), suggesting that afferent input from extensor muscles is suppressed more during the flexion trials than during the extension trials. Such movement direction-related modulation was also reported in previous studies using the wrist extension-flexion task, e.g. the reciprocal activity of premotor interneurons in the spinal cord²⁹ and the reciprocal activity of neurons in the primary motor area³⁰.

Then, we examined the polarity of the directional modulation of PAD. In this analysis, we exclusively used the static task epochs (i.e. the Delay and Active Hold [AH] epochs), which comprised the largest part of each trial (75–82%) and exhibited a consistent feature within each movement direction (Fig. 2d, thin dotted and dashed lines). The dynamic task epochs were excluded because we had already identified direction-independent modulation during these epochs.”

Comment 1-2.

And one semantic comment: using the phrase that a test did not "reach" statistical significance (twice in this manuscript, including Fig 2c) implies that the authors think that there is a difference between the two groups of data, but that, perhaps, there's not enough power (or some other explanation). But that is not what the data are objectively saying: the data are telling us that we do not have the statistical grounds to reject the null hypothesis that the two groups are the same.

Answer 1-2.

We agree with this comment. We have changed the wording to a more objective manner as shown in lines 131–137 (line 130 in the previous version) and lines 288–289 (line 247 in the previous version), saying:

“In the wrist flexion trials, it was noteworthy that the sole negative mean was observed in the AM epoch, in common with the wrist extension trials, although none of the epochs differed from the Rest period (Delay, mean = 0.26, $t = 1.44$, $p = 0.13$; AM, mean = -0.03, $t = 0.21$, $p = 0.83$; AH, mean = 0.33, $t = 1.95$, $p = 0.05$; PM, mean = 0.04, $t = 0.26$, $p = 0.79$, $df = 76$, p -values are uncorrected, paired two-tailed t -test with Bonferroni’s correction, correction size = 4).”

and

“we found no significant correlation between ADV size and the success ratio (Fig. 5f, $df = 76$, $t = 1.63$, $p = 0.11$), but...,”

Reviewer #2 (Remarks to the Author):

In this manuscript, the authors explore how proprioceptive feedback may be modulated at first-order spinal interneurons by pre-synaptic inhibition. The authors have done an excellent job of addressing my previous concerns with the manuscript. However, there are a few additional concerns I would like the authors to address.

Comment 2-1

A major finding of the present study was that PAD modulation emerged prior to a motor action suggesting a contribution from a descending cortical pathway. Although I don't disagree that this pathway plays a key role, I find the evidence supporting this claim is rather weak. One piece of evidence is that ICA analysis revealed components (IC2) that exhibited PAD modulation in the delay period that was direction specific. However, it's unclear whether these components are genuinely meaningful or whether they are simply the dimensionality technique trying to find structure in the noise. This result would be better bolstered if the manuscript included a statistical analysis to demonstrate that the results are not spurious. At the very least the ICs could include confidence intervals by bootstrapping the means of each ADV across trials. This method can also provide a probability value by asking how many times did the bootstrap distributions for the extension and flexion movements show the expected result (i.e. for IC2 flexion > 0 and extension trials <0). However, I leave it to the authors to determine the best method of statistically verifying their ICA results.

Answer 2-1

Thank you very much for this comment and providing an idea that can be used to reinforce our argument about the contribution of a descending cortical pathway on PAD modulation. As advised, we now added the confidence intervals for each epoch of IC1–4 (Fig. 3a), which were obtained by using bootstrapped data, i.e. the ICA of 1,000-times resampling the actual dataset with replacement. We found that most values, including IC2, exceeded the 95% confidence interval, suggesting that they are independent of noise and thus meaningful.

Moreover, according to your advice, we also obtained 1,000 medians of mixing weights for the bootstrap dataset and estimated their 95% confidence intervals. If the bootstrap data showed the expected result, the actual weights would be inside the confidence intervals, but they were not. Therefore, we can say that the difference between movement directions in our actual data was not spurious. To reflect these changes, we modified Fig. 3a and the corresponding description in the Results section (lines 197–199 and 657–660) and the legend of Fig. 3 (lines 1021–1027):

“The IC values and medians of mixing weights exceeded the 95% confidence interval (Fig. 3a and 3b, dark grey lines), suggesting they are independent of noise and, thus, significant. “

“To obtain the confidence intervals for ICA, we generated a 1,000-times bootstrap dataset with replacement from the actual dataset (154 × 4 matrix), and performed ICA. The obtained confidence intervals of ICs and mixing weights were illustrated as dark grey lines (Fig. 3a–b).”

“a. Four independent components (ICs) for the behavioural modulation of antidromic volleys (ADV)s. Black bars indicate confidence intervals. AH, Active Hold; AM, Active

Movement; PM, Passive Return Movement. b. Median value of the weight of each IC (mixing weight, $n = 77$). Black bars indicate confidence intervals. c. Presumed pattern of ADV modulation, illustrated based on the extracted ICs (a) and mixing weights (b). d. Example of four ADVs exhibiting pattern modulation of their size across different behavioural epoch and their waveforms (inset).”

Comment 2-2

In a similar vein, the manuscript’s interpretation of Figure 5 is a bit incorrect. In particular, Figure 5e-g shows that PAD modulation can correlate with trial success during the static hold epoch. However, the authors interpret Figure 5 in the discussion as (line 333) “In contrast, during agonistic movements, we found that successful trials were preceded by larger PSI (Fig. 5).” (see also line 251 in the Results). This interpretation is rather problematic as the data actually reflect the pooling of the Delay and Active Hold epochs (line 228, i.e. static hold epoch). As such, it is unclear to the reader whether this correlation truly reflects modulation in the Delay epoch or might be largely generated by the Active Hold epoch. The manuscript would benefit from not pooling the Delay and Active Hold periods and showing the results for each epoch separately.

Answer 2-2

In the previous version, the statement in line 333 was regarding the extension trials (agonistic to wrist extensor) and it described Fig. 5a–d, not Fig. 5e–g. This analysis used the PAD in the AM epoch, not the Delay and AH epochs. Still, we found the description “successful trials were preceded by larger PSI” confusing since the AM epoch is a part of the trial. Therefore, we changed the wording to (lines 384–385):

”In contrast, during agonistic movements, we found that the successful trials exhibited larger PSI in the period with dynamic motor output (i.e. AM epoch, Fig. 5).”

As for the analysis that combined the Delay and AH periods, our intention was to represent the entire active trials without including dynamic movements in which the pattern of ADV modulation was clearly different, as now stated more explicitly in the revised manuscript (lines 152–157, 269–270, and 379):

“Then, we examined the polarity of the directional modulation of PAD. In this analysis, we exclusively used the static task epochs (i.e. the Delay and Active Hold [AH] epochs), which comprised the largest part of each trial (75–82%) and exhibited a consistent feature within each movement direction (Fig. 2d, thin dotted and dashed lines). The dynamic task epochs were excluded because we had already identified direction-independent modulation during these epochs.”

“a static task epoch (combination of Delay and AH; assessing comprehensive modulation in each trial,...”

“the larger PSI throughout the flexion trials (Fig. 2) was...”

We agree with the comment that separate analysis would be beneficial because we could address the behavioural consequence of the descending modulation (together with Fig. 3) of PAD. Therefore, we performed this analysis separately for the Delay and AH epochs, as shown below. The results confirmed the observation from the combined dataset (Fig. 5f in the main text), showing no significant difference in the success rate between the lower and higher ADV trials. However, we could not confirm the significant difference in the amplitude of flexor EMG activity between the small and large ADV trials against the combined results (see circles in Fig. 5g), although the similar profiles (relatively larger EMG amplitude in the trials with larger ADVs) seemed to be maintained in both the Delay and AH epochs.

The results of this auxiliary analysis suggest that the findings for the combined dataset do not exclusively reflect the AH period, as the reviewer expressed concern about. However, we feel this result was not conclusive for two reasons. First, the size of modulation in the Delay and AH epochs during flexion movements was moderate (Fig. 2), compared with that in the AM epoch during extension movements. Second, the trial-based analysis necessarily has a

weaker signal-to-noise ratio. Therefore, this analysis might not be sensitive enough to detect subtle changes in ADV size differentially in the Delay and AH epochs, and differentially according to task performance.

Therefore, we decided not to include this figure in the revised manuscript and to keep the original figure. Instead, we introduced the results and implication of this analysis (lines 292–298):

“Since this result was not replicated when we repeated a comparable analysis of the ADVs elicited in the Delay and AH epochs separately (Delay, $df = 76$, $t = 1.23$, $p = 0.22$; AH, $df = 76$, $t = 0.53$, $p = 0.60$, two-tailed paired t -tests), we could not ascribe this observation to the unique features of either the Delay or AH epoch. Rather, this result represents a characteristic of the flexion trials and suggests that the larger PAD throughout a trial helps to sustain EMG activity. Overall, these results...”

As suggested in Comment 2-3, this result may not be directly coherent with the IC2 profile shown in Fig. 2. We acknowledged this point in the revised manuscript (see Answer 2-3).

Comment 2-3

The manuscript should acknowledge the apparent contradiction presented throughout the paper about whether ADV modulation is present in the delay period. The results presented in Figure 2C suggest no statistically significant modulation in the Delay period for either the extension or flexion movements (though a small positive bias exists for the flexion movement). Similarly, Figure 4A indicates significant ADV modulation starting 100ms after EMG onset which is after the Delay period. In contrast, results in Figure 3 and 5 tell a different story with significant modulation in the Delay period. I recognize that Figure 2C and Figure 4A reflect trial and ADV averages that likely remove finer modulation effects that are more readily observed using ICA analysis. It would be a useful point to acknowledge this point and may be a useful additional motivation in the Results for the ICA analysis.

Answer 2-3.

We agree with the reviewer that the reader may be confused by the contradictory results between Figs. 2 and 4 (no significant modulation in the Delay period) and Fig. 3 (IC2 showed modulation in the Delay period; for our argument about the Delay period modulation in Fig. 5). We also agreed with the reviewer’s suggestion that *“Figure 2C and Figure 4A reflect trial and ADV averages that likely remove finer modulation effects that are more readily observed using ICA analysis.”*

Therefore, we addressed this comment as follows.

1) In the revision, we now clearly state the motivation to perform ICA (Fig. 3) in addition to the analysis of epoch averages (Fig. 2) (lines 189–193):

“These sources might not be represented in the total average (Fig. 2) because they could operate PAD in parallel during voluntary movements and thus their effect might be offset to each other. To compensate for this disadvantage, we extracted the components underlying the observed task-dependent modulation by independent component analysis (ICA)...”

2) We described the implication of the contradiction shown in the total averages in Figs. 2c and 4a and the ICA analysis for IC1 and IC2 (lines 247–252):

“While the characteristics of the AM epoch in IC1 were also reflected in the results of averaging analyses (Figs. 2c and 4a, significant difference with the Rest period), those of the Delay epoch in IC2 were less represented (Figs. 2c and 4a, no significant difference with the Rest period). These results suggest that a larger population of ADVs is modulated by the descending source for IC1, but a smaller subpopulation is affected by the descending source for IC2.”

3) We also described a potential cause of the contradiction in the ADV modulation in the Delay period (lines 347–351):

“ICA suggested that this reciprocal modulation of PSI in the flexion and extension tasks was already observed in the Delay period, i.e. a period for preparing future movements (IC2 in Fig. 3). This reciprocal modulation during the Delay period was less reflected in simple analysis using averaged data (Figs. 2c and 4a), suggesting it may represent the characteristics of a smaller subpopulation of ADVs.”

Comment 2-4

Minor points

Caption for Figure 5 requires significantly more detail. Also, the layout of Figure 5 causes problems. I realize now the red-blue color scheme denotes flexion and extension movements however, with the way the figure is laid out I had initially assumed it corresponded to the AM epoch and Static Hold epoch based on sub panels A and E. This initially confused me when trying to interpret sub panels D and H. Thus, The figure should present the data from the same direction in A and E to prevent confusion.

Answer 2-4.

In this revision, we modified Fig. 5a to avoid any confusion. For example, we used the same direction for the description of the analysed time window (Fig. 5a and 5e in the previous version, lines 268-270). We also added more information to the legend for this figure (lines 1043–1058):

“a dynamic task epoch (300 ms from EMG onset; shading in Fig. 5a upper panel) and a static task epoch (combination of Delay and AH; assessing comprehensive modulation in each trial, shading in Fig. 5a lower panel).”

*“a. Assessment windows (shaded areas) for computing the mean ADV area for each trial. b. Sample error (black) and successful (grey) trials of wrist torque and EMG during extension movements. c. Sample error (black) and successful (grey) trials of wrist torque and EMG during flexion movements. d. Hold success ratio in the extension trials classified by the mean ADV in the dynamic task epoch. *p < 0.05 (t-test). e. Means and standard errors of the EMG amplitude of individual wrist extensor and flexor muscles during the Active Hold (AH) epoch of extension trials classified by the mean ADV in the dynamic task epoch. Circle, p < 0.05 for t-test between large and small ADV trials. APL, abductor pollicis longus; BRD, brachioradialis; ECR, extensor carpi radialis; ECU, extensor carpi ulnaris; ED23, extensor digitorum-2,3; ED45, extensor digitorum-4,5; EDC, extensor digitorum communis; FCR, flexor carpi radialis; FCU, flexor carpi ulnaris; FDS, flexor digitorum superficialis; PL, palmaris longus; PT, pronator teres. f. Same as for panel d, but the data were from flexion trials classified by the mean ADV in the static task epoch. g. Same as for panel e, but the data were from flexion trials classified by the mean ADV in the static task epoch. d–g are also illustrated in Supplementary Figure 3.”*

Comment 2-5

Currently, the manuscript only describes that ADV shape correlates with the amount of PAD. Can the authors be more specific by including at the beginning of the Results that the size of the ADV shape positively correlates with the amount of PAD.

Answer 2-5.

We added text to the Results section in accordance with this comment (lines 115–119):

“The response to each microstimulation during wrist movements was normalized to those observed in the canonical period (Rest), compiled independently for each behavioural epoch, and then used to evaluate changes in PAD, as the level of PAD positively correlated with the size of ADVs elicited by intraspinal stimulation²¹.”

Comment 2-6

In the Results, reporting of t-tests should also include an explicit statement of the degrees of freedom and the size of the Bonferroni correction factor that was used.

Answer 2-6.

We added statistical information in accordance with this comment. For example (lines 124–127):

“In the wrist extension trials, PAD suppression occurred during the Active Movement (AM) epoch (Fig. 2c, mean = -0.44, df = 76, t = 3.06, uncorrected p = 0.003, paired two-tailed t-test with Bonferroni’s correction, correction size = 4, compared with Rest),...”

This information was also added to lines 129–131, 134–136, 145, 159–164, 170–172, 280, 285–286, 288–291, 294–295, and 309–316.

Comment 2-7

Can the author’s include more details about the kinematic and EMG filters (i.e. type and poles, Lines 516 and 521).

Answer 2-7.

We added filter information in accordance with this comment (lines 541–544):

“Data were amplified and filtered by MCP-Plus (Alpha Omega, Nazareth Illit, Israel; high-pass filter = 5 Hz for EMG and 500 Hz for ADV, two-pole Butterworth filter; low-pass filter = 3 kHz for EMG and 10 kHz for ADV, four-pole Butterworth filter)....”

(lines 545–548):

“...or amplified, filtered, and digitized by AlphaLabSnR (Alpha Omega, Nazareth Illit, Israel; pre-amplification hardware high-pass filter = 0.5 Hz, two-pole Butterworth filter; pre-amplification hardware low-pass filter = 10 kHz, three-pole Butterworth filter) for Monkey Y.”

Reviewer #3 (Remarks to the Author):

The authors have satisfactorily addressed my comments.

REVIEWERS' COMMENTS

Reviewer #2 (Remarks to the Author):

The authors have satisfied all my issues with the manuscript.

REVIEWERS' COMMENTS

Reviewer #2 (Remarks to the Author):

The authors have satisfied all my issues with the manuscript.

Our answer:

We are sincerely grateful to the reviewer for providing numerous constructive, insightful, and valuable comments throughout the revision process. These comments have enabled us to explore our data from new perspectives and gain a deeper understanding of its characteristics. As a result, we firmly believe that our manuscript has become more valuable.